# SYMMETRY-DRIVEN DISCOVERY OF DYNAMICAL VARIABLES IN MOLECULAR SIMULATIONS

## ABSTRACT

Molecular dynamics simulations are crucial for understanding complex biomolecular systems, but they are often hindered by the high dimensionality of the configurational space. This paper introduces two novel approaches for discovering effective degrees of freedom (DoF) in molecular dynamics simulations by leveraging approximate symmetries of the energy landscape. We present a scalable symmetry loss function compatible with existing force-field frameworks and a Hessian-based method efficient for smaller systems. Both approaches enable systematic exploration of conformational space by connecting structural dynamics to energy landscape symmetries. Applied to alanine dipeptide, our methods comprehensively sample the Ramachandran plot, including shallow minima. Simulations initiated from our DoF-sampled points converge to all important conformations, demonstrating the methods' effectiveness in navigating complex energy landscapes. These approaches offer powerful tools for efficient exploration in molecular simulations, with potential applications in protein folding and drug discovery.

## 1 INTRODUCTION

Molecular dynamics (MD) is an essential tool for a wide range of applications, including drug discovery, protein folding, and understanding the physics of biological systems at the molecular level . By simulating the motion of atoms and molecules over time, MD allows researchers to investigate the structural and functional properties of complex biomolecules and materials. The configuration space of molecules, however, is extremely high-dimensional, with each atom contributing three degrees of freedom (xyz coordinates). Despite this complexity, the physically relevant conformations of molecules typically occupy a much lower-dimensional subspace. This space corresponds to regions of low free energy, which can be thought of as a negative log-likelihood of the system's state. The most likely conformations correspond to the minima of this energy landscape.

Sampling from this low-energy subspace is crucial for understanding the function and stability of biomolecules, but it presents significant computational challenges due to the vast number of possible configurations and the presence of energy barriers separating different conformational states. Existing methods for sampling these low-energy conformations include enhanced sampling techniques such as metadynamics Laio & Parrinello (2002), umbrella sampling Torrie & Valleau (1977), and replica exchange molecular dynamics (REMD) Sugita & Okamoto (1999), which aim to overcome energy barriers and improve exploration of the conformational space. However, these methods often require careful tuning and can be computationally expensive.

In this work, we present a novel approach for discovering degrees of freedom (DoF) that effectively move the system along the low-energy manifold, enabling more efficient exploration of relevant conformations in the molecular landscape. Our key observation is that low-energy DoF (LEDoF) can be related to approximate symmetries of the energy function. Recent works in machine learning have made progress in discovering symmetries in data Benton et al. (2020); Dehmamy et al. (2021); Yang et al. (2023b;a). Our setting is different in that we do not have data, but an energy function, which can be used to construct a symmetry, loss. This problem was addressed in LieGG Moskalev et al. (2022), where we discover infinitesimal symmetry generators, i.e. Lie algebra elements.

We find approximate symmetries by considering small transformations of the original DoF and deriving conditions for the near-invariance of the energy. This process yields a symmetry loss, which

we then minimize. In the case of small molecules, we show that the problem can be formulated as finding symmetries of the Hessian. Aside from the optimization approach, we also provide an analytic solution based on degenerate Hessian eigenspaces.

Our method connects the structural dynamics of molecules to the approximate symmetries of the energy landscape. This enables a systematic exploration of conformational space that is both computationally efficient and physically insightful. We implement this technique in the study of Alanine dipeptide and demonstrate how these effective DoFs can correspond to significant chemical features such as dihedral angles. Hence, these DoFs facilitate targeted exploration and sampling of biologically relevant conformations. WE show that this method of sampling can significantly reduce the computational burden associated with high-dimensional MD simulations. It also enhances the ability to explore areas of the conformational landscape that are difficult to access through conventional methods.

Our contributions can be summarized as:

1. **DoF as Symmetries:** We formulate identifying effective DoF (EDoF) as an optimization problem aiming to discover approximate symmetries of the energy.

2. **Symmetry discovery using the Hessian:** We derive the relation between symmetry generators and the Hessian near critical points. We also provide a method to construct symmetries from the spectrum of the Hessian.

3. **Extraction of internal coordinates for molecules:** We show that our method leads to discovering well-known dihedral angles and several additional EDoF.

4. **Effective exploration of Alanine dipeptide landscape:** We demonstrate enhanced capability in sampling diverse conformations, particularly shallow local minima, which are difficult to sample using existing methods.

## 2 RELATED WORK

Exploring the multidimensional energy landscape of molecular systems, such as proteins or molecules, poses significant challenges due to the system's numerous degrees of freedom, which make the landscape rugged. Traditional molecular dynamics and Monte Carlo methods often struggle to fully map this landscape, as they tend to become trapped in local minima and fail to capture rare conformations. This is largely due to the high energy barriers surrounding the minima or the long timescales required for conformational changes to evolve Bernardi et al. (2015).

**Space sampling.** Monte Carlo methods are advantageous for sampling configurational space due to the absence of inherent timescales, but they struggle to capture transitions between conformations and can become trapped in local minima behind high-energy barriers, leaving some regions of the energy landscape poorly sampled Heilmann et al. (2020). Umbrella sampling, introduced by Torrie & Valleau (1977), addresses this by replacing the standard Boltzmann weighting with a biasing potential, effectively enabling a random walk across energy barriers.

Methods such as replica exchange molecular dynamics (REMD) Hansmann & Okamoto (1999); Sugita & Okamoto (1999) employ molecular dynamics simulations simultaneously on a series of replica systems with different conditions,e.g. temperature, and randomly exchange the states of any two replicas with a regular schedule Qi et al. (2018). By utilizing multiple replicas, REMD enables efficient sampling of the energy landscape and helps overcome high energy barriers. However, the need for numerous replicas can significantly increase computational demands, making the approach challenging to implement in practice Rathore et al. (2005); Liu et al. (2005); Wang et al. (2020).

Metadynamics is another approach to improve sampling of the energy landscape of a system by driving it through collective variables (CV), which represent key coordinates defining the landscape Laio & Parrinello (2002); Bussi & Laio (2020). In molecular dynamics simulations, Gaussian biases are periodically added to prevent the system from revisiting previously explored regions, facilitating the discovery of new minima. In many cases, identifying the appropriate set of collective variables (CVs) is non-trivial and typically relies on prior knowledge; additionally, managing a large number of CVs can become inefficient.

Coarse graining and machine learning methods can also accelerate molecular dynamics simulations, improving sampling efficiency and expanding accessible phase space Souza et al. (2021); Majewski et al. (2023); Noé et al. (2020).

**Identifying the DoF.** Identifying suitable collective variables (CVs) for energy landscape sampling is challenging, often introducing biases and facing issues related to data quality and interpretability. Principal component analysis (PCA) has been employed to define CVs Hori et al. (2009), effectively capturing broad folding and non-folding features of proteins. However, due to its linear nature, PCA often fails to represent the intricate and nonlinear structure of protein energy landscapes Maisuradze et al. (2009). To address these limitations, autoencoders have been used to automatically learn nonlinear CVs from data Chen & Ferguson (2018), although this approach may yield CVs that are difficult to interpret physically and dependent on data quality. Support vector machines (SVMs) Sultan & Pande (2018) have also been applied to classify protein properties from short molecular dynamics simulations and to identify CVs. Other machine learning techniques, such as self-supervised deep neural networks, have been developed to identify slow CVs or reaction coordinates, further advancing the search for effective descriptors Wehmeyer & Noé (2018).

## 3 THEORY

This section presents our theoretical framework for discovering effective degrees of freedom (DoF) in molecular systems, based on identifying approximate symmetries in the energy landscape. We introduce two approaches: a scalable symmetry loss function compatible with existing force-field frameworks, and a Hessian-based method effective for smaller systems. Both methods connect molecular structural dynamics to energy function symmetries, enabling systematic conformational space exploration. We derive the mathematical foundations of these approaches, showing how they lead to the discovery of physically meaningful DoF, such as dihedral angles in peptides.

### 3.1 SYMMETRY AND DOF

While we primarily focus on physical and molecular systems, our approach is general and can be formulated in broader terms by treating the energy as a loss function. Let $E : \mathcal{Z} \to \mathbb{R}$ represent the potential energy of a physical system, which, analogous to a loss function, we assume to be smooth over large regions of the parameter space $\mathcal{X}$, and bounded from below. The parameters $z \in \mathcal{Z}$ correspond to the system's degrees of freedom (DoF). We assume that $\mathcal{X}$ is a vector space. In the context of molecular dynamics (MD), the standard DoF include the 3D positions $\vec{x}_i$ and momenta $\vec{p}_i = m_i d\vec{x}_i/dt = m_i \dot{\vec{x}}_i$ of all particles $i \in \{1, \ldots, n\}$, where $m_i$ denotes the mass of particle $i$.

**Temperature and kinetic energy** In this work, we are primarily interested in the static conformations of the system, and thus, we ignore the kinetic energy term in our formulation. By focusing on the potential energy, we capture the equilibrium properties of molecular systems. While temperature induces thermal fluctuations in real systems, our current approach neglects these effects. However, these fluctuations could be incorporated in future extensions, particularly when accounting for finite-temperature effects and exploring dynamic behavior. Ignoring the kinetic DoF $p_i$, our parameter space reduces to the space $\mathcal{X} \subset \mathcal{X}$ of positions $x = \{\vec{x}_i\}$. Hence, we redefine the energy to be just the **potential energy** $E : \mathcal{X} \to \mathbb{R}$

**Lifting DoF to a group** Our core idea is to use transformations acting on the degrees of freedom (DoF) instead of the original DoF, thereby lifting the DoF to a group action on $\mathcal{X}$. While not all DoF can be lifted in this manner, we demonstrate that this approach enables us to link low-energy DoF to underlying symmetries of the system. We consider the general linear group $GL(\mathcal{X})$ acting on the parameter space $\mathcal{X}$. The translation group can also be included, but since MD potentials are generally translation invariant, it leads to trivial symmetries, which we are not interested in here. Starting from a reference point $x_0 \in \mathcal{X}$, the orbit of $GL(\mathcal{X})$, defined as $\text{Orbit}\{x = gx_0 \mid g \in GL(\mathcal{X})\}$, generates a manifold of transformed configurations. This manifold effectively describes the set of configurations related to $x_0$ by symmetry transformations. This allows us to replace $x$ with transformations $g$ that reach $x$ from $x_0$. By focusing on symmetries that preserve or approximately preserve the potential energy, we extract DoF that correspond to motion along low-energy directions, thus providing a natural way to explore the low-energy landscape of the system.

**Group parameters as DoF** In order to map $g$ to degrees of freedom, we need a parametrization for $g$. Since $GL(\mathcal{X})$ is a continuous group, we use the Lie algebra and the exponential map to parametrize $g$ in terms of the Lie algebra basis $\boldsymbol{L}_a \in \mathfrak{gl}(\mathcal{X})$, where the Lie algebra $\mathfrak{gl}(\mathcal{X}) = T_{\mathrm{id}}GL(\mathcal{X})$ is the tangent space at the identity of $GL(\mathcal{X})$. In the case of matrix Lie groups such as $GL(\mathcal{X})$, the exponential map $\exp : \mathfrak{gl}(\mathcal{X}) \to GL(\mathcal{X})$ can be written in terms of the matrix exponential. Exponentiating an element in the Lie algebra yields a group element $g = \exp(\theta \cdot \boldsymbol{L})$, where $\theta$ is a vector of parameters. In more general cases, where a group element requires a nontrivial path on the group manifold, the transformation may be expressed as a product of exponentials $g = \prod_i \exp(\theta_i \cdot \boldsymbol{L})$. In both cases, small group elements (i.e. near identity) can be expanded as $g \approx \boldsymbol{I} + \theta \cdot \boldsymbol{L} + O(\theta^2)$, where $\boldsymbol{I}$ is the identity matrix. This formulation allows us to define the group parameters $\theta$ as the new DoF, reparametrizing the system in terms of group transformations that capture the low-energy dynamics. Next, we define more concretely what we mean by low-energy dynamics and effective DoF.

## 3.2 Effective Degrees of Freedom

To motivate our definition of effective degrees of freedom (DoF), we begin by considering the potential energy $E(z)$ of a molecular system described by a typical force field, such as AMBER Cornell et al. (1995). The energy landscape comprises quadratic terms representing bond lengths and bond angles, which can be minimized easily, along with non-linear terms such as Lennard-Jones and Coulomb forces that dominate at short distances and decay at larger separations (Appendix A).

Once the bond and angle terms have been minimized, the system navigates a restricted subspace of the full configuration space where bond lengths and angles remain near their equilibrium values. However, within this subspace, the energy can still vary significantly due to non-linear interactions between atoms. To describe the relevant motion in this subspace, we seek directions in which the potential energy changes minimally. These directions correspond to the effective DoF of the system.

Concretely, let $x \in \mathcal{X}$ be a reference point, corresponding to a configuration where the quadratic energy terms are minimized. Consider a small perturbation $\delta x$, such that the system moves from $x$ to $x' = x + \delta x$. The first-order expansion of the energy around $x$ is $E(x + \delta x) \approx E(x) + \delta x \cdot \nabla E(x)$. We are interested in directions $\delta x$ for which $\delta x \cdot \nabla E(x)$ is small, implying minimal change in the energy. Now, instead of arbitrary perturbations, we focus on transformations generated by elements of the Lie algebra. Let $g \approx I + \epsilon \boldsymbol{L}$ be a small group transformation acting on $x$, so that $x = gx$ and the perturbation is $\delta x = \boldsymbol{L}x$. Substituting into the energy expansion, we have

$$E(gx) \approx E(x) + \boldsymbol{L}x \cdot \nabla E(x) \tag{1}$$

We define the **effective DoF** as the directions in the Lie algebra for which the directional derivative $Lx \cdot \nabla E(x)$ is small. These directions correspond to approximate symmetries of the system, in the sense that they preserve the energy to first order. Next, we discuss how to learn these DoF.

## 3.3 Symmetry loss

To uncover approximate symmetries in the system, we introduce a symmetry loss function that penalizes deviations in the energy under small transformations. Specifically, for a transformation $g$ acting on a reference configuration $x \in \mathcal{X}$, we seek to minimize the difference in potential energy before and after the transformation, such that $|E(gx) - E(x)| < \eta$ for some small threshold $\eta$. As discussed above, we parametrize the group elements as $g = \exp(\theta \cdot \boldsymbol{L})$ using the Lie algebra basis $\boldsymbol{L}_a$, making each $\theta_a$ a new DoF. Thus, to identify the DoF we need to find the condition satisfied by the Lie algebra $\boldsymbol{L}_a$. To do so, we consider a small transformations $g \approx I + \epsilon \boldsymbol{L}$, where $\epsilon$ is a small parameter. Substituting this into the energy function and expanding to first order, we obtain the following condition for $L$

$$E(gx) \approx E(x) + \epsilon \nabla E(x) \cdot \boldsymbol{L}x \tag{2}$$

Thus, the symmetry loss translates to $|\epsilon \nabla E(x) \cdot \boldsymbol{L}x|$. To get a more well-behaved function under gradient-based optimization, we use $(\epsilon \nabla E(x) \cdot \boldsymbol{L}x)^2$ instead. The corresponding optimization objective is to find the Lie algebra element $\boldsymbol{L}$ that minimizes the symmetry loss

$$\textbf{Symmetry loss:} \quad \mathcal{L}(\boldsymbol{L}, x) = (\nabla E(x) \cdot \boldsymbol{L}x)^2 \tag{3}$$

subject to $\|\boldsymbol{L}\|_F \leq 1$, ensuring that $\boldsymbol{L}$ remains within a bounded region of the Lie algebra. This optimization allows us to identify approximate symmetries that correspond to low-energy directions in the configuration space, which can then be used to define effective DoF.

This formulation closely parallels the approach used in LieGG Moskalev et al. (2022), where Lie group generators are learned in a similar fashion to uncover symmetries in learned representations. By minimizing this symmetry loss, we systematically identify transformations in the Lie algebra that correspond to low-energy directions in the configuration space, thereby linking approximate symmetries to the physically meaningful degrees of freedom of the system. However, in problems such as MD there are important global symmetry considerations, which we discuss next.

### 3.4 EXCLUDING GLOBAL SYMMETRIES

In MD, the configuration space $\mathcal{X}$ is naturally isomorphic to $\mathbb{R}^{n \times d}$, where $n$ is the number of particles and $d$ is the spatial dimension. For $d = 3$, the system often has a global $SE(3)$ symmetry, corresponding to rotations and translations in three-dimensional space. However, we are not interested in this symmetry, as it represents trivial motions that do not affect the relative configuration of the particles. Therefore, we restrict the Lie algebra element $\boldsymbol{L}$ to act on the particle indices, while being invariant under $SE(3)$ transformations.

Given this restriction, the action of $\boldsymbol{L}$ affects only the $n$-dimensional part of $x \in \mathbb{R}^{n \times d}$. The condition for approximate symmetry, $(\nabla E \cdot Lx)^2$, can now be written as

$$SE(3)\text{-invariant loss:} \quad \mathcal{L}(\boldsymbol{L}, x) = \left( \sum_{i,j,\mu} \frac{\partial E}{\partial x_j^\mu} \boldsymbol{L}_j^i x_i^\mu \right)^2 = \left( \mathrm{Tr} \left[ (\nabla E)^\top \boldsymbol{L} x \right] \right)^2 \tag{4}$$

where $i, j$ index the particles, and $\mu$ indexes the spatial components. Here, $\nabla E \in \mathbb{R}^{n \times d}$ is the gradient of the energy with respect to the particle positions, and the matrix product involves $\boldsymbol{L}$, which acts on the particle indices, while $x$ is the current configuration of the system. We will be working with equation 4 instead of equation 3. Additionally, in small molecular systems we can use another level of simplification using the Hessian, described next.

### 3.5 HESSIAN APPROACH FOR SYMMETRY LOSS

Assume that $x_*$ is a critical point of the energy, meaning that $\nabla E(x_*) = 0$. Let $x = x_* + \epsilon$, where $\epsilon$ is a small perturbation around $x_*$. We can Taylor expand $\nabla E(x)$ around $x_*$ to first order

$$\nabla E(x) \approx \nabla E(x_*) + H(x_*) \cdot (x - x_*) = H(x_*) \cdot \epsilon \tag{5}$$

Here, $H(x_*)$ is the Hessian matrix at $x_*$, which has components $H_{\mu\nu}^{ij} = \partial^2 E / \partial x_i^\mu \partial x_j^\nu$.

**Expectation over Gaussian Perturbations** Now, assume that we have many samples $x$, such that $\epsilon$ is drawn from a Gaussian distribution. We take the expectation of the above expression. First, Note that the cross-term $\mathbb{E}\left[ \mathrm{Tr} \left[ H \boldsymbol{L} x_* \epsilon^\top \right] \right] = 0$, vanishes because $\mathbb{E}[\epsilon] = 0$ The second term, involving $\epsilon^\top H \boldsymbol{L} \epsilon$, remains. Since $\epsilon \sim \mathcal{N}(0, \sigma^2 I)$, we have

$$\mathbb{E}[\epsilon_i^\mu \epsilon_k^\nu] = \sigma^2 \delta_{ik} \delta^{\mu\nu} \tag{6}$$

Substituting this expansion into the symmetry loss

$$\mathbb{E}[\mathcal{L}(\boldsymbol{L}, x)] = \mathbb{E}\left[ \left( \mathrm{Tr} \left[ \epsilon^\top H(x_*)^\top \boldsymbol{L}(x_* + \epsilon) \right] \right)^2 \right] \tag{7}$$

where $H = H(x_*)$ and we used the symmetry of the Hessian ($H^\top = H$). The cross-term in the square is $O(\epsilon^3)$ and vanishes because $\epsilon$ is normal. Since $H$ and $\boldsymbol{L}$ always appear together and for ease of notation, let us denote $\boldsymbol{K} \equiv H\boldsymbol{L}$. For the first term in equation 7, using equation 6 we get (see Appendix A.1 for all derivations below)

$$\mathbb{E}\left[ \mathrm{Tr} \left[ \epsilon^\top \boldsymbol{K} x_* \right]^2 \right] = \sigma^2 \|\boldsymbol{K} x_*\|^2 \tag{8}$$

The last term, this in equation 7 yields

$$\mathbb{E}\left[\operatorname{Tr}\left[\epsilon^\top \boldsymbol{K}\epsilon\right]^2\right] = \sigma^4 \left\{2\operatorname{Tr}\left[\boldsymbol{K}_S^2\right] + \operatorname{Tr}\left[\boldsymbol{K}_S\right]^2\right\} \tag{9}$$

where $\boldsymbol{K}_S = (\boldsymbol{K}^\top + \boldsymbol{K})/2$ is the symmetric part of $\boldsymbol{K}$. Putting these together, the symmetry loss in this approximation becomes

**Hessian symmetry loss:**

$$\mathbb{E}\left[\mathcal{L}(\boldsymbol{L}, x)\right] \approx \sigma^2 \|\boldsymbol{K}x_*\|^2 + \sigma^4 \left\{2\operatorname{Tr}\left[\boldsymbol{K}_S^2\right] + \operatorname{Tr}\left[\boldsymbol{K}_S\right]^2\right\}. \tag{10}$$

### 3.6 Analytical Solutions to the Trace Loss in 1D

We note that there exists a simple analytical ways to minimize each of the two loss terms.

**Minimizing $O(\sigma^2)$ term:** This can be done as follows:

**Proposition 3.1** ($\boldsymbol{L}$ in null space of the Hessian). *If $\boldsymbol{L}$ has support only on the approximate null space of $\boldsymbol{H}_2 := \sum_{\mu\nu} H_{\mu\nu}^2$ then it also minimizes $\operatorname{Tr}\left[\boldsymbol{K}x_*\right]^2$ can be minimized.*

*Proof.* One way $\operatorname{Tr}\left[\boldsymbol{K}x_*\right]^2 = \|H\boldsymbol{L}x_*\|^2$ can be minimized is if $\boldsymbol{L}$ satisfies:

$$\forall \mu\nu: \quad H_{\mu\nu}\boldsymbol{L} \approx 0. \tag{11}$$

This can be achieved by minimizing the sum of squares of all terms in equation 11, which yields

$$\|H\boldsymbol{L}\|^2 = \operatorname{Tr}\left[\boldsymbol{L}^T \boldsymbol{H}_2 \boldsymbol{L}\right], \quad \boldsymbol{H}_2 := \sum_{\mu\nu} H_{\mu\nu}^2. \tag{12}$$

In turn, $\operatorname{Tr}\left[\boldsymbol{L}^T \boldsymbol{H}_2 \boldsymbol{L}\right]$ is minimized if $\boldsymbol{L}$ has support only on the null space of $\boldsymbol{H}_2$. $\qquad\square$

In practice, we are content with having small but nonzero $\operatorname{Tr}\left[\boldsymbol{L}^T \boldsymbol{H}_2 \boldsymbol{L}\right]$. In this case, $\boldsymbol{L}$ corresponds to directions in the configuration space along which the energy is approximately flat.

**2. Minimizing $O(\sigma^4)$ only:** These terms only depend on the symmetric part of $\boldsymbol{K} = H\boldsymbol{L}$ and:

**Proposition 3.2** (Anti-symmetric $\boldsymbol{K}$). *If $H\boldsymbol{L}$ is antisymmetric, the trace loss can also be minimized. This requires that*

$$H\boldsymbol{L} + \boldsymbol{L}^\top H = 0 \quad \Rightarrow \quad \operatorname{Tr}\left[\boldsymbol{K}_S\right] = 0, \quad \operatorname{Tr}\left[\boldsymbol{K}_S^2\right] = 0$$

*which implies that $\boldsymbol{L}$ generates transformations that preserve the structure of the Hessian. One solution to this condition is if $\boldsymbol{L}$ itself is antisymmetric $\boldsymbol{L}^\top = -\boldsymbol{L}$. In this case, the commutator between $\boldsymbol{H}$ and $\boldsymbol{L}$ vanishes*

$$[\boldsymbol{L}, H] = 0$$

*This implies that $\boldsymbol{L}$ commutes with $H$, and therefore defines symmetry directions where the Hessian is invariant.*

If $H$ has a degenerate subspace corresponding to $k$ degenerate eigenvalues, this subspace has an inherent $SO(k)$ symmetry. Because of this, the Lie algebra elements $\boldsymbol{L} \in \mathfrak{so}(k)$ of this subspace symmetry satisfy $[\boldsymbol{L}, H] = 0$. This is a special case of the proposition 3.2. More formally:

**Proposition 3.3** (Degenerate subspace solution). *let $\Lambda$ be the diagonalized form of $H_{\mu\nu}$, with $Q\Lambda Q^\top = H_{\mu\nu}$. If $\Lambda$ has a set of $k$-fold degenerate eigenvalues $\lambda_1 = \lambda_2 = \cdots = \lambda_k$, the corresponding eigenspace forms a $k$-dimensional subspace of symmetry. The action of $\boldsymbol{L}$ in this subspace can be viewed as a rotation, and $\boldsymbol{L}$ can be chosen to belong to the Lie algebra of rotations $SO(k)$ restricted to the degenerate subspace:*

$$\boldsymbol{L} \in \mathfrak{so}(k), \quad \boldsymbol{L}^\top = -\boldsymbol{L} \tag{13}$$

*The matrix $\boldsymbol{L}$ generates rotations within the degenerate eigenspace, leaving the overall structure of $H_{\mu\nu}$ invariant.*

In our experiments, because of the global spatial symmetries, instead of looking for degenerate eigenspaces of for each $H_{\mu\nu}$, we instead look at the spectrum of the $SE(3)$ invariant matrix $\boldsymbol{H}_2$ in equation 12, which further unifies the proposed solutions to the $\sigma^2$ and $\sigma^4$ terms in the symmetry loss equation 10. .

This mechanism applies specifically to the case where $\boldsymbol{H}_2$ has degenerate eigenvalues, as it is the degeneracy that gives rise to the $SO(k)$ symmetry. If the eigenvalues are non-degenerate, such a rotation within an eigenspace does not apply, and the symmetry must be realized in other ways, such as by ensuring $\boldsymbol{L}$ lies in the approximate null space of $\boldsymbol{H}_2$ or satisfies an antisymmetry condition.

## 4 SELECTING OPTIMAL EFFECTIVE DOF

Using the discovery procedure, we get a collection of Lie Algebra Elements corresponding to the degrees of freedom close to a local minima in the energy landscape. Given a collection of Lie Algebra elements, we need to find the transformations that help us navigate the free energy landscape most effectively. There might be multiple small degrees of freedom corresponding to small rotations of the hydrogen atoms in a free methyl group while some others might lead to rotations around peptide bonds causing much larger structural changes in the protein.

In this section, we want to solve the problem of finding the most effective DOF's given a collection of symmetries. The effective-ness of a symmetry corresponds to the magnitude of change in structure that results from using the symmetry. In order to optimize for effectiveness we quantify it as the generators that lead to the largest perturbation in energy at very small noise levels. Using the result from equation 8 as a proxy of the magnitude of structural change, we can solve the following optimization problem to get the most effective DoFs:

$$\boldsymbol{L}(a) := \sum_{i=1}^{n_L} a_i \boldsymbol{L}_i, \qquad \text{Find } a^* := \underset{a \in \mathbb{R}^K, \|a\|_2 = 1}{\arg\max} \|H\boldsymbol{L}(a)x_*\|^2 \qquad (14)$$

where $n_L = k(k-1)/2$ is the dimensions of the Lie algebra $\mathfrak{so}(k)$ in a degenerate subspace $H(x^*)$, and $x^*$ is a local minima. Solving this quadratic equation in $n_L$ variables we can get the top most effective DoFs from our collection.

For the optimization version of the problem, we see that as mentioned before the Lie algebra generators are discovered at high noise levels by minimizing the symmetry loss over the samples. In order to find the most effective degrees of freedom, we try to maximize the symmetry loss function at low noise levels in order to remove any vibrational DoFs.

Given $m$ values of $\epsilon_j \sim \mathcal{N}(0, \sigma_{\textbf{eff}}\mathbf{I}^{n \times d})$ where $\sigma_{\textbf{eff}}$ is smaller than $\sigma$ used for discovery. Then

$$a^* := \underset{a \in \mathbb{R}^K, \|a\|_2 = 1}{\arg\max} \sum_{j=1}^{m} \text{Tr} \left[ \nabla E(x_* + \epsilon_j)^\top \boldsymbol{L}(a)\epsilon_j \right]^2 \qquad (15)$$

where $m$ is the number samples $x = x_* + \epsilon_i$. We apply this methodology to finding effective DoF in Alanine dipeptide, as described next.

## 5 EXPERIMENTS

We apply our method for finding effective degrees of freedom on Alanine Dipeptide. In the case of alanine-dipeptide, the two most effective degrees of freedom are known to be the $\phi$ and $\psi$ dihedral angles over the peptide bonds. Our goal is to re-discover these degrees of freedom from the forcefield without using any prior knowledge about the importance of the peptide bonds. We start all our experiments at the $\beta$-sheet conformation for Alanine Dipepetide and compare the discovered conformers and the explored regions of the space wrt the conformers discovered from running long simulations. For each experiment we provide the labelled conformers discovered in the simulation and also a density plot highlighting the areas of the Ramachandran plot covered by our simulations.

As the conformers of alanine dipeptide as well as the ramachandran plot change with the ambient medium, we consider two MD setting: 1) **Alanine Dipepetide in Vacuum** where we only use the amber forcefield (amber99sbnmr) corresponding to interactions within the molecule and 2) **Alanine**

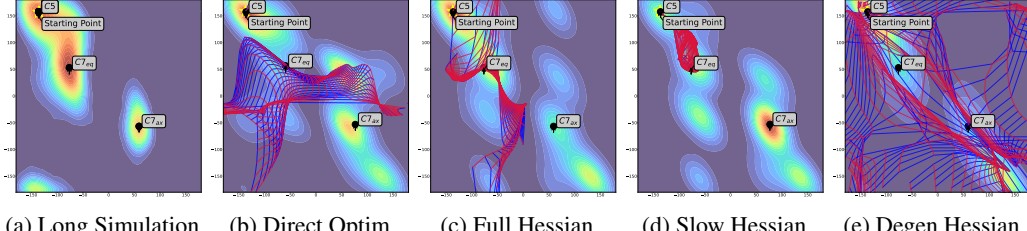

(a) Long Simulation     (b) Direct Optim     (c) Full Hessian     (d) Slow Hessian     (e) Degen Hessian

Figure 1: Ramachandran plot for alanine dipeptide in vacuum based on the a) long (500 ns) simulation starting at $\beta$ alanine dipeptide, b) direct otimization based approach, c) analytically solving the full $\sigma^4$ term in equation 3 d) slow subspace of $\boldsymbol{H}_2$ e) fast degenerate subspace of $\boldsymbol{H}_2$. The blue and red grid lines on the plots refer to the grid traced by transforming $\beta$ alanine using the two most effective DOF discovered by our algorithms.

**Dipeptide in Water** where we use the molecule forcefield (amber99sbnmr) along with the amber forcefield for the solvent (amber99_obc) modelled as implicit.

For our baseline we run two long openMM simulations (one with water and one without any solvent) for $500ns$ at $300K$ with friction coefficient of $1ps^{-1}$ and step size of $2fs$ amounting to $2.5e8$ steps. We use the amber forcefields in both the last step of our method and the baseline simulations in order to maintain consistency. For each of the settings considered, we provide results from all three approaches considered in this paper:

1. Hessian-based approach solving equation 12 using the Eigenvectors of the $\boldsymbol{H}_2$ matrix. This yields multiple solutions: one corresponding to each slow subspace and one for each degenerate subspace spanned by the eigenvectors of $\boldsymbol{H}_2$. We only present the solution from the slow subspace and from the biggest degenerate subspace.

2. Solving the Symmetry Loss equation corresponding to the $\sigma^4$ term in equation 3 without making any additional simplifications for discovering the symmetries.

3. Directly solving the optimization version of the Symmetry loss that does not require Hessian computation and only uses molecule positions sampled by perturbing the molecule around a local minima with varying levels of Gaussian noise. While the Hessian-based methods do not easily generalize to the setting with a stochastic solvent force, the optimization version is also used to solve the system with the implicit solvent model. We discuss the results for both settings : optimization method using just intra-molecular forces and optimization method including solvent forcefield.

In the experiments presented in this section, we use the two most effective symmetry directions generated by our method to conduct a grid search over the combination of the symmetries. We use a grid of $31 \times 31$ points to form the starting points for running openmm simulations. For each starting point on the grid formed by our method, we start an openmm simulation by calling the minimize function on the atom positions and then running a $2ps$ (1000 steps) simulation using the exact same parameters as used for the baseline simulation.

## 5.1 EXPERIMENTS IN VACUUM

When modelling Alanine dipeptide in vacuum, we only consider the inter-molecular forces between the atoms. For this setting, we see that almost all the Hessian-based methods recover all the major conformations of alanine dipeptide with relatively short simulation times. Although the optimization based approach models the problem with the fewest assumptions, we see that it sometimes has poorer performance due to the higher variance stemming from its stochastic nature. As the Lie algebra generators considered in our problem span a $\mathbb{R}^{n^2}$ space, we need at least $O(n^2)$ point to avoid overfitting. For our experiments, we use $5n^2$ samples for discovery and $5n^2$ samples for finding the most effective degrees of freedom. Using larger values of $\epsilon$ can give us more information about long-range symmetries but this also increase the stochasticity causing very high variance in the estimates.

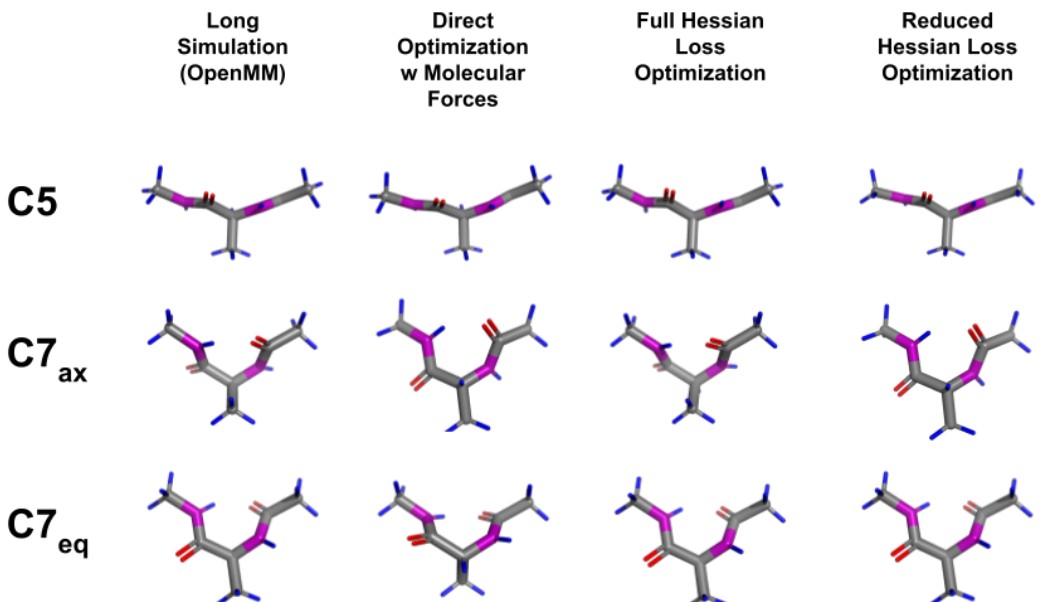

Figure 2: Closest Structure to the conformers of Alanine Dipeptide Discovered by our method for simulations in Vacuum. The blank space correspond to undiscovered conformers. We only consider a structure to be a conformer candidate if it is stable and it's $\phi$ and $\psi$ dihedral angles are close to that of the corresponding conformer.

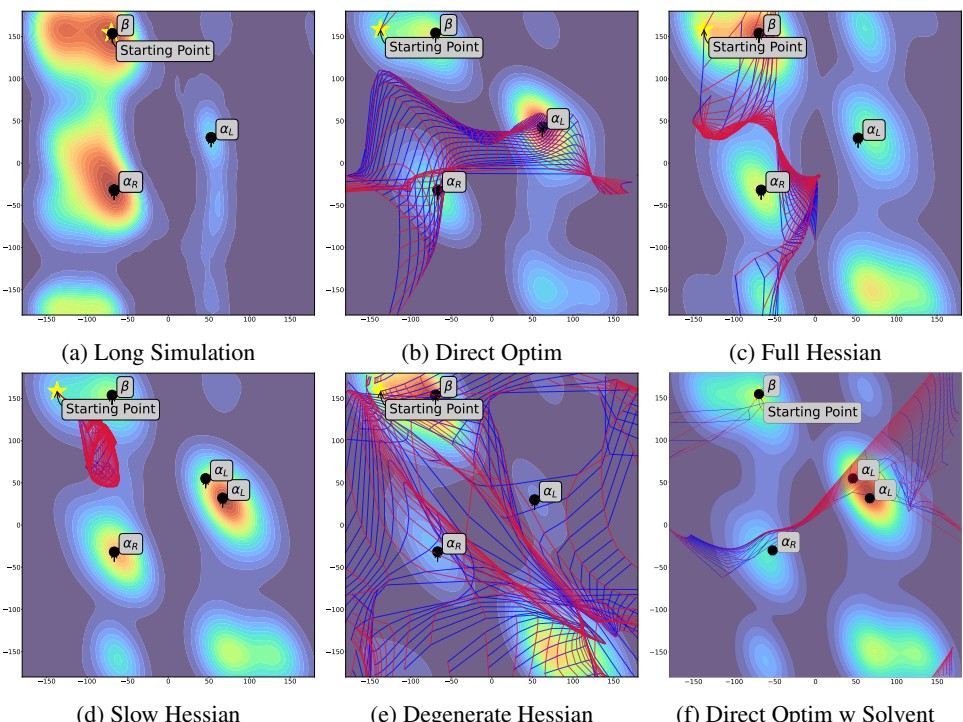

Figure 3: Ramachandran plot for alanine dipeptide in water based on the a) long (500 ns) openMM simulation with implicit solvent starting at $\beta$ alanine dipeptide, b) direct optimization based approach over the molecular forcefield, c) analytically solving the full $\sigma^4$ term in equation 3 d) slow subspace of $H_2$ e) fast degenerate subspace of $H_2$ f) direct optimization based approach over the solvent and the molecular forcefield . The blue and red grid lines on the plots refer to the grid traced by transforming $\beta$ alanine using the two most effective DOF discovered by our algorithms.

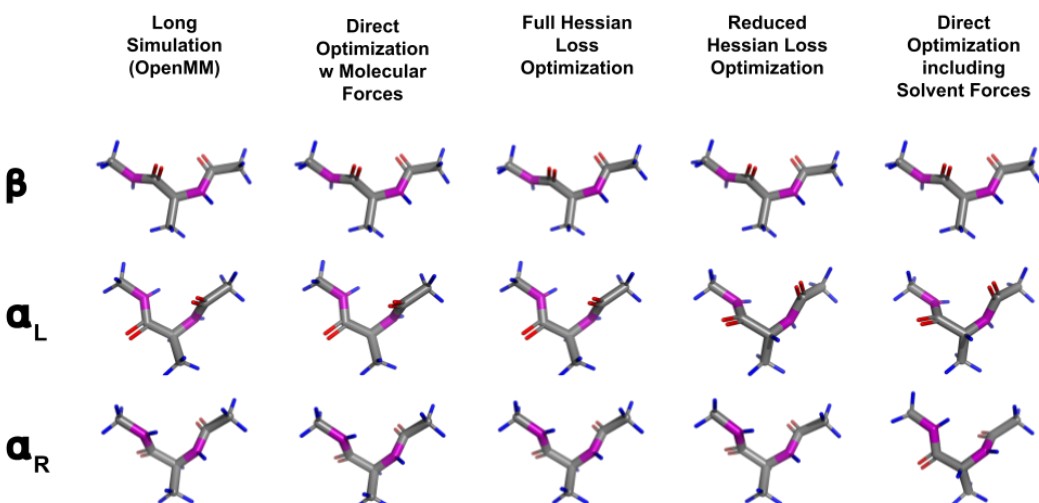

Figure 4: Closest Structure to the conformers of Alanine Dipeptide Discovered by our method for simulations in Water. For water finding conformers in water, we require our discovered symmetries to generalize as the Hessian computation does not directly work for solvent-like forces. Only the algorithm in the last column uses the solvent force field to calculate the DoFs.

## 5.2 EXPERIMENTS WITH SOLVENT

Alanine Dipeptide has different stable conformers in water which shows that the knowledge of the solvent force is very important in order to model the symmetries. However, we see that the degrees of freedom still remain invariant across the two case. So, we use our the DOFs discovered in the vacuum to also navigate the free energy landscape in the presence of solvent.

Moreover, we also see that although the Hessian-based approached do not easily transfer over to the solvent scenario, the optimization based approach can be easily generalized to any description of a forcefield. So, we also incorporate the implicit solvent forcefield in the optimization based approach to find symmetries over the combined forcefields. As seen from the results above, almost all the approaches proposed discover the major conformers of alanine dipeptide under both the settings. Another important fact that we want to highlight is that discovered DOFs are general enough to allow us to find the conformers is settings with forcefields slightly different from the ones they were learnt on.

**Discussion about the time advantage** For both the Hessian-based approached, we see that we are able to fully recover the main conformations of alanine dipeptide while also sampling other saddle points or unstable conformations. Although our method uses the openmm simulation in the final step, we only need a fraction of the total steps required by openmm to find all conformations. Furthermore our methods does not need to be run sequentially as the simulations for all the grid points can be run simultaneously. Thus, in principle, the effective time required for simulating our method could be $10^5 \times$ less than the time required to get the same results using openmm simulations. The major bottleneck for our approach is the Hessian computation which scales quadratically with the size of the system and does not easily generalize to stochastic forcefields. While the optimization method overcomes the second challenge in principle we find it to be less stable than the Hessian-based approach in practice. As for the quadratic cost of our method, it inherently comes from modelling the pairwise interactions between the particles and it can be pruned to be almost linear by using cut-offs parameters for intra-atom interactions. We plan to explore this further in future work.

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

## A AMBER FORCE-FIELD

We implement a simplified force-field with implicit solvent (i.e. water molecules are not modeled and appear as hydrogen-bonding and hydrophobicity terms. In protein folding our energy function consists of five potential energies for: bond length $E_{bond}$, bond angles $E_{angle}$, Van der Waals $E_{vdW}$, hydrophobic $E_{hp}$ and hydrogen bonding $E_H$ Ceci et al. (2007).

**Protein Folding with Classical MD Using AMBER Force Field** We incorporate the AMBER force field, known for its accurate representation of molecular interactions, particularly in proteins. This force field is implemented using the parameters from OpenMM Eastman et al. (2017), and it comprehensively models the following interactions:

- Bond lengths $E_{bond}$ and bond angles $E_{angle}$
- Torsional angles $E_{torsion}$
- Non-bonded interactions including van der Waals $E_{vdW}$ and electrostatic $E_{elec}$ forces

We utilize the functional forms and parameters specified in the AMBER force field:

$$E_{bond} = \sum_{bonds} k_{bond}(r - r_0)^2 \qquad E_{angle} = \sum_{angles} k_{angle}(\theta - \theta_0)^2 \qquad (16)$$

$$E_{torsion} = \sum_{torsions} V_n \left[1 + \cos(n\omega - \gamma)\right] \qquad E_{vdW} = \sum_{i<j} \frac{A_{ij}}{r_{ij}^{12}} - \frac{B_{ij}}{r_{ij}^6} \qquad (17)$$

$$E_{elec} = \sum_{i<j} \frac{q_i q_j}{4\pi\epsilon_0\epsilon_r r_{ij}} \qquad (18)$$

Here, $r$ and $\theta$ represent the bond lengths and angles, respectively, with $r_0$ and $\theta_0$ as their equilibrium values. The torsional term $E_{torsion}$ includes a sum over all torsion angles $\omega$, with periodicity $n$, amplitude $V_n$, and phase $\gamma$. The Lennard-Jones potential in $E_{vdW}$ is characterized by parameters $A_{ij}$ and $B_{ij}$, and $E_{elec}$ is calculated using the Coulombic potential with partial charges $q_i$, $q_j$ and the relative permittivity $\epsilon_r$.

In this simulation, we exclude the modeling of solvent effects entirely, focusing solely on the protein in vacuum. This approach simplifies the computational model while emphasizing the direct interactions within the protein.

The overall energy of the system is then given by:

$$\mathcal{L}(X) = E_{bond} + E_{angle} + E_{torsion} + E_{vdW} + E_{elec} \qquad (19)$$

### A.1 HESSIAN SYMMETRY LOSS

Substituting this expansion into the symmetry loss

$$\mathbb{E}[\mathcal{L}(\boldsymbol{L}, x)] = \mathbb{E}\left[\left(\text{Tr}\left[\epsilon^\top H(x_*)^\top \boldsymbol{L}(x_* + \epsilon)\right]\right)^2\right]$$

$$= \mathbb{E}\left[\text{Tr}\left[\epsilon^\top H\boldsymbol{L}x_*\right]^2 + 2\text{Tr}\left[\epsilon^\top H\boldsymbol{L}x_*\right]\text{Tr}\left[\epsilon^\top H\boldsymbol{L}\epsilon\right] + \text{Tr}\left[\epsilon^\top H\boldsymbol{L}\epsilon\right]^2\right] \qquad (20)$$

where $H = H(x_*)$ and we used the symmetry of the Hessian ($H^\top = H$). For the second term we have

$$\mathbb{E}\left[\text{Tr}\left[\epsilon^\top H\boldsymbol{L}x_*\right]\text{Tr}\left[\epsilon^\top H\boldsymbol{L}\epsilon\right]\right] = 0 \quad (\text{vanishes due to } \mathbb{E}[\epsilon^3] = 0) \qquad (21)$$

Since $H$ and $\boldsymbol{L}$ always appear together and for ease of notation, let us denote $\boldsymbol{K} \equiv H\boldsymbol{L}$. For the first term in equation 20, using equation 6 we get

$$\mathbb{E}\left[\text{Tr}\left[\epsilon^\top \boldsymbol{K}x_*\right]^2\right] = \sum_{i,j,\mu,\nu} \mathbb{E}\left[\epsilon_i^\mu \epsilon_j^\nu\right](\boldsymbol{K}x_*)_\mu^i (\boldsymbol{K}x_*)_\nu^j \qquad (22)$$

$$= \sigma^2 \sum_{i,\mu}(\boldsymbol{K}x_*)_\mu^i (\boldsymbol{K}x_*)_\mu^i = \sigma^2 \|\boldsymbol{K}x_*\|^2 \qquad (23)$$

The last term, this in equation 20 yields

$$\mathbb{E}\left[\mathrm{Tr}\left[\epsilon^\top \boldsymbol{K}\epsilon\right]^2\right] = \sum_{i,j,k,l,\mu,\nu,\rho,\lambda} \boldsymbol{K}_{\mu\nu}^{ij}\boldsymbol{K}_{\rho\lambda}^{kl}\mathbb{E}\left[\epsilon_i^\mu \epsilon_j^\nu \epsilon_k^\rho \epsilon_l^\lambda\right] \tag{24}$$

$$= \sigma^4 \sum_{i,j,\mu,\nu}\left\{\boldsymbol{K}_{\mu\nu}^{ij}\left[\boldsymbol{K}_{\mu\nu}^{ij} + \boldsymbol{K}_{\nu\mu}^{ji}\right] + \boldsymbol{K}_{\mu\mu}^{ii}\boldsymbol{K}_{\nu\nu}^{jj}\right\} \tag{25}$$

$$= \sigma^4\left\{\mathrm{Tr}\left[\boldsymbol{K}\left(\boldsymbol{K}^\top + \boldsymbol{K}\right)\right] + \mathrm{Tr}\left[\boldsymbol{K}\right]^2\right\}$$

$$= \sigma^4\left\{\frac{1}{2}\mathrm{Tr}\left[\left(\boldsymbol{K}^\top + \boldsymbol{K}\right)^2\right] + \mathrm{Tr}\left[\boldsymbol{K}\right]^2\right\} \tag{26}$$

Defining the symmetric part $\boldsymbol{K}_S = (\boldsymbol{K}^\top + \boldsymbol{K})/2$, we have

$$\mathbb{E}\left[\mathrm{Tr}\left[\epsilon^\top \boldsymbol{K}\epsilon\right]^2\right] = \sigma^4\left\{2\mathrm{Tr}\left[\boldsymbol{K}_S^2\right] + \mathrm{Tr}\left[\boldsymbol{K}_S\right]^2\right\} \tag{27}$$

where we used the fact that $\mathrm{Tr}\left[\boldsymbol{K}\right] = \mathrm{Tr}\left[\boldsymbol{K}_S\right]$. Putting these together, the symmetry loss in this approximation becomes

**Hessian symmetry loss:**

$$\mathbb{E}\left[\mathcal{L}(L, x)\right] \approx \sigma^2\|\boldsymbol{K}x_*\|^2 + \sigma^4\left\{2\mathrm{Tr}\left[\boldsymbol{K}_S^2\right] + \mathrm{Tr}\left[\boldsymbol{K}_S\right]^2\right\}. \tag{28}$$

Now, note that since $\boldsymbol{K} \equiv H\boldsymbol{L}$ the components are $\boldsymbol{K}_{\mu\nu}^{ij} = \sum_k H_{\mu\nu}^{ik}\boldsymbol{L}_k^j$. Let us define the trace over spatial indices, $\mu, \nu$, and the node (particle) indices $i, j$ as follows

$$\mathrm{Tr_s}\left[H\right]^{ij} \equiv \sum_\mu H_{\mu\mu}^{ij}, \qquad\qquad \mathrm{Tr_n}\left[H\right]_{\mu\nu} \equiv \sum_i H_{\mu\nu}^{ii} \tag{29}$$

For the trace terms we have

$$\mathrm{Tr}\left[\boldsymbol{K}\right] = \sum_{i,\mu}\boldsymbol{K}_{\mu\mu}^{ii} = \sum_{i\mu}H_{\mu\mu}^{ik}\boldsymbol{L}_k^i = \mathrm{Tr_n}\left[\mathrm{Tr_s}\left[H\right]\boldsymbol{L}\right]$$

$$\mathrm{Tr}\left[\boldsymbol{K}\boldsymbol{K}^\top\right] = \sum_{ij\mu\mu}\left(\boldsymbol{K}_{\mu\nu}^{ij}\right)^2 = \sum_{i\mu}H_{\mu\nu}^{ik}\boldsymbol{L}_k^j H_{\mu\nu}^{il}\boldsymbol{L}_l^j = \mathrm{Tr_n}\left[\boldsymbol{L}^\top \mathrm{Tr_s}\left[H^2\right]\boldsymbol{L}\right]$$

$$\mathrm{Tr}\left[\boldsymbol{K}^2\right] = \sum_{ij\mu\mu}\left(\boldsymbol{K}_{\mu\nu}^{ij}\right)^2 = \sum_{i\mu}H_{\mu\nu}^{ik}\boldsymbol{L}_k^j H_{\nu\mu}^{jl}\boldsymbol{L}_l^i = \mathrm{Tr_s}\left[\mathrm{Tr_n}\left[H\boldsymbol{L}H\boldsymbol{L}\right]\right] \tag{30}$$

This provides a relationship between the Hessian, the Lie algebra element $\boldsymbol{L}$, and the effective DoF defined by $\boldsymbol{L}$, allowing us to identify approximate symmetries by minimizing this loss. Note that the above still approximately holds even $x_*$ is not a critical point, but some point where the gradient is small, meaning $|\nabla E(x_*)| < \eta$ for some small $\eta$.

# B DERIVATIONS

## B.1 HESSIAN SYMMETRY LOSS

For the first term in equation 7, using equation 6 we get

$$\mathbb{E}\left[\mathrm{Tr}\left[\epsilon^\top H\boldsymbol{L}x_*\right]^2\right] = \sum_{i,j,\mu,\nu}\mathbb{E}\left[\epsilon_i^\mu \epsilon_j^\nu\right](H\boldsymbol{L}x_*)_\mu^i(H\boldsymbol{L}x_*)_\nu^j \tag{31}$$

$$= \sigma^2 \sum_{i,\mu}(H\boldsymbol{L}x_*)_\mu^i(H\boldsymbol{L}x_*)_\mu^i = \|H\boldsymbol{L}x_*\|^2 \tag{32}$$

Finally, the last term in equation 7 yields

$$\mathbb{E}\left[\mathrm{Tr}\left[\epsilon^\top H\boldsymbol{L}\epsilon\right]^2\right] = \sum_{i,j,k,l,\mu,\nu,\rho,\lambda}(H\boldsymbol{L})_{\mu\nu}^{ij}(H\boldsymbol{L})_{\rho\lambda}^{kl}\mathbb{E}\left[\epsilon_i^\mu \epsilon_j^\nu \epsilon_k^\rho \epsilon_l^\lambda\right] \tag{33}$$

$$= \sigma^4 \sum_{i,j,\mu,\nu}\left\{(H\boldsymbol{L})_{\mu\nu}^{ij}\left[(H\boldsymbol{L})_{\mu\nu}^{ij} + (H\boldsymbol{L})_{\nu\mu}^{ji}\right] + (H\boldsymbol{L})_{\mu\mu}^{ii}(H\boldsymbol{L})_{\nu\nu}^{jj}\right\} \tag{34}$$

## B.2 ANALYTICAL SOLUTIONS TO THE TRACE LOSS IN 1D

We can make this term vanish by finding $\boldsymbol{L}$ that satisfies

$$\text{if: } \boldsymbol{L}^\top H + H\boldsymbol{L} = 0 \quad \Rightarrow \quad \text{Tr}\left[\boldsymbol{K}_S^2\right] = 0. \tag{35}$$

A special case of this is when $L$ is anti-symmetric, $\boldsymbol{L} = -\boldsymbol{L}^\top$. This amounts to $\boldsymbol{L}$ generating a *rotation in the particle space*. In this case, equation 35 becomes $[H, \boldsymbol{L}] = 0$, meaning **if a rotation commutes with** $H$, it yields $\text{Tr}\left[\boldsymbol{K}_S^2\right] = 0$. Note that $H$ has both particle $ij$ and spatial indices $\mu\nu$. The exact commutation condition is

$$\forall(\mu, \nu), \quad \text{if: } [\boldsymbol{L}, H_{\mu\nu}] = 0 \quad \Rightarrow \quad \text{Tr}\left[\boldsymbol{K}_S^2\right] = 0. \tag{36}$$

. We can find even more explicit solutions to the commutation relation. On simple solution to $[H, \boldsymbol{L}] = 0$ can be found when $H$ has a *degenerate* subspace, meaning when two or more eigenvalues are the same. In this subspace, $H$ is proportional to identity. Therefore, any $\boldsymbol{L}$ that has only support on a degenerate subspace commutes with $H$. In our experiments, because of the global spatial symmetries, instead of looking for degenerate eigenspaces of for each $H_{\mu\nu}$, we instead look at the spectrum of the $SE(3)$ invariant matrix $\boldsymbol{H} \equiv \sum_\mu \nu [HH^T]_{\mu\nu}$ .

We are interested in solving the trace loss:

$$\mathcal{L}_{\text{Hess}}(x) = \sigma^2 \left| \text{tr}(\boldsymbol{H}(x)\boldsymbol{L}) \right|$$

where $\boldsymbol{H} \equiv \sum_\mu H_{\mu\mu}$ is the spatially traced Hessian and $\boldsymbol{L}$ is the Lie algebra element defining the transformation.

We propose different ways to satisfy the trace loss analytically:

Now let's consider potential additional ways to satisfy the trace condition analytically:

**Proposition B.1.** *If $\boldsymbol{L}$ is a block matrix with specific structure that aligns with the eigenbasis of $\boldsymbol{H}$, we can also minimize the loss. Specifically, if we diagonalize $\boldsymbol{H}$ as $\boldsymbol{H} = Q\Lambda Q^\top$, where $\Lambda$ is diagonal, then by choosing $\boldsymbol{L}$ such that it aligns with this eigenbasis, we can ensure that:*

$$\text{Tr}\left[\boldsymbol{H}\boldsymbol{L}\right] = \text{Tr}\left[Q\Lambda Q^\top \boldsymbol{L}\right]$$

*If $\boldsymbol{L}$ is aligned such that it only has support in the eigenspaces corresponding to small eigenvalues of $\Lambda$, the trace can be minimized. This approach effectively identifies directions in which the Hessian has minimal influence.*

**Proposition B.2.** *Another possible solution is if $\boldsymbol{L}$ has a low-rank structure, particularly if it is a rank-one or rank-two matrix. In this case, even if $\boldsymbol{H}$ has high rank, the product $\boldsymbol{H}\boldsymbol{L}$ will still result in a matrix of low rank, potentially minimizing the trace loss:*

$$\text{Tr}\left[\boldsymbol{H}\boldsymbol{L}\right] = \sum_i \lambda_i v_i^\top \boldsymbol{L} v_i$$

*where $v_i$ are the eigenvectors of $\boldsymbol{H}$ and $\lambda_i$ are the corresponding eigenvalues. If $\boldsymbol{L}$ is chosen to have support in directions orthogonal to the eigenvectors corresponding to large eigenvalues, the trace loss can be reduced.*

In the case where $\boldsymbol{H}$ has degenerate eigenvalues, an additional symmetry arises. Specifically, if $\boldsymbol{H}$ has a degenerate subspace corresponding to $k$ degenerate eigenvalues, this subspace has an inherent $SO(k)$ symmetry. In such cases, we can choose $\boldsymbol{L}$ to be a generator of rotations within the degenerate subspace, exploiting the symmetry. More formally:

**Proposition B.3.** *let $\Lambda$ be the diagonalized form of $\boldsymbol{H}$, with $Q\Lambda Q^\top = \boldsymbol{H}$. If $\Lambda$ has a set of $k$-fold degenerate eigenvalues $\lambda_1 = \lambda_2 = \cdots = \lambda_k$, the corresponding eigenspace forms a $k$-dimensional subspace of symmetry. The action of $\boldsymbol{L}$ in this subspace can be viewed as a rotation, and $\boldsymbol{L}$ can be chosen to belong to the Lie algebra of $SO(k)$, the group of rotations in $k$ dimensions.*

*In this case, $\boldsymbol{L}$ takes the form of an antisymmetric matrix restricted to the degenerate subspace:*

$$\boldsymbol{L} \in \mathfrak{so}(k), \quad \boldsymbol{L}^\top = -\boldsymbol{L}$$

*The matrix $\boldsymbol{L}$ generates rotations within the degenerate eigenspace, leaving the overall structure of $\boldsymbol{H}$ invariant. This rotation symmetry, arising due to the degeneracy of the eigenvalues, ensures that the trace loss $\mathrm{tr}(\boldsymbol{H}\boldsymbol{L})$ vanishes. Thus, the commutator condition:*

$$[\boldsymbol{L}, \boldsymbol{H}] = 0$$

*is naturally satisfied within this degenerate subspace. The presence of degenerate eigenvalues gives rise to this additional symmetry, and $\boldsymbol{L}$ can be chosen as one of the canonical generators of $SO(k)$.*

This mechanism applies specifically to the case where $\boldsymbol{H}$ has degenerate eigenvalues, as it is the degeneracy that gives rise to the $SO(k)$ symmetry. If the eigenvalues are non-degenerate, such a rotation within an eigenspace does not apply, and the symmetry must be realized in other ways, such as by ensuring $\boldsymbol{L}$ lies in the approximate null space of $\boldsymbol{H}$ or satisfies an antisymmetry condition.

## C   FURTHER EXPERIMENTAL RESULTS

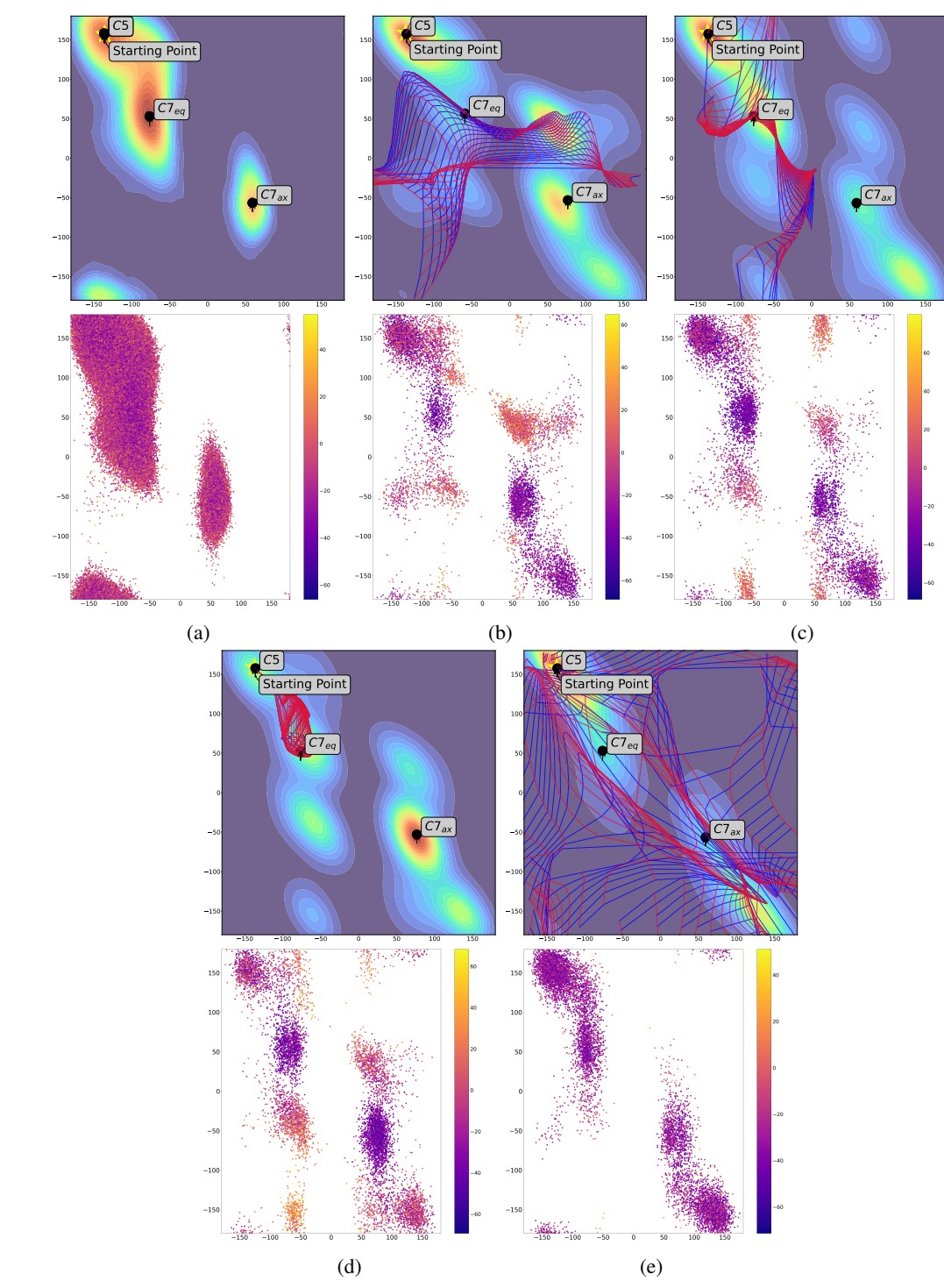

Figure 5: Ramachandran plot for alanine dipeptide in Vacuum based on the a) long (500 ns) openMM simulation with implicit solvent starting at $\beta$ alanine dipeptide, b) direct optimization based approach over the molecular forcefield, c) analytically solving the full $\sigma^4$ term in equation 3 d) slow subspace of $H_2$ e) fast degenerate subspace of $H_2$ f) direct optimization based approach over the solvent and the molecular forcefield . The blue and red grid lines on the plots refer to the grid traced by transforming $\beta$ alanine using the two most effective DOF discovered by our algorithms. Additionally the scatter plot gives the values of the potential energy (in presence of solvent) at the points sampled using the corresponding method.

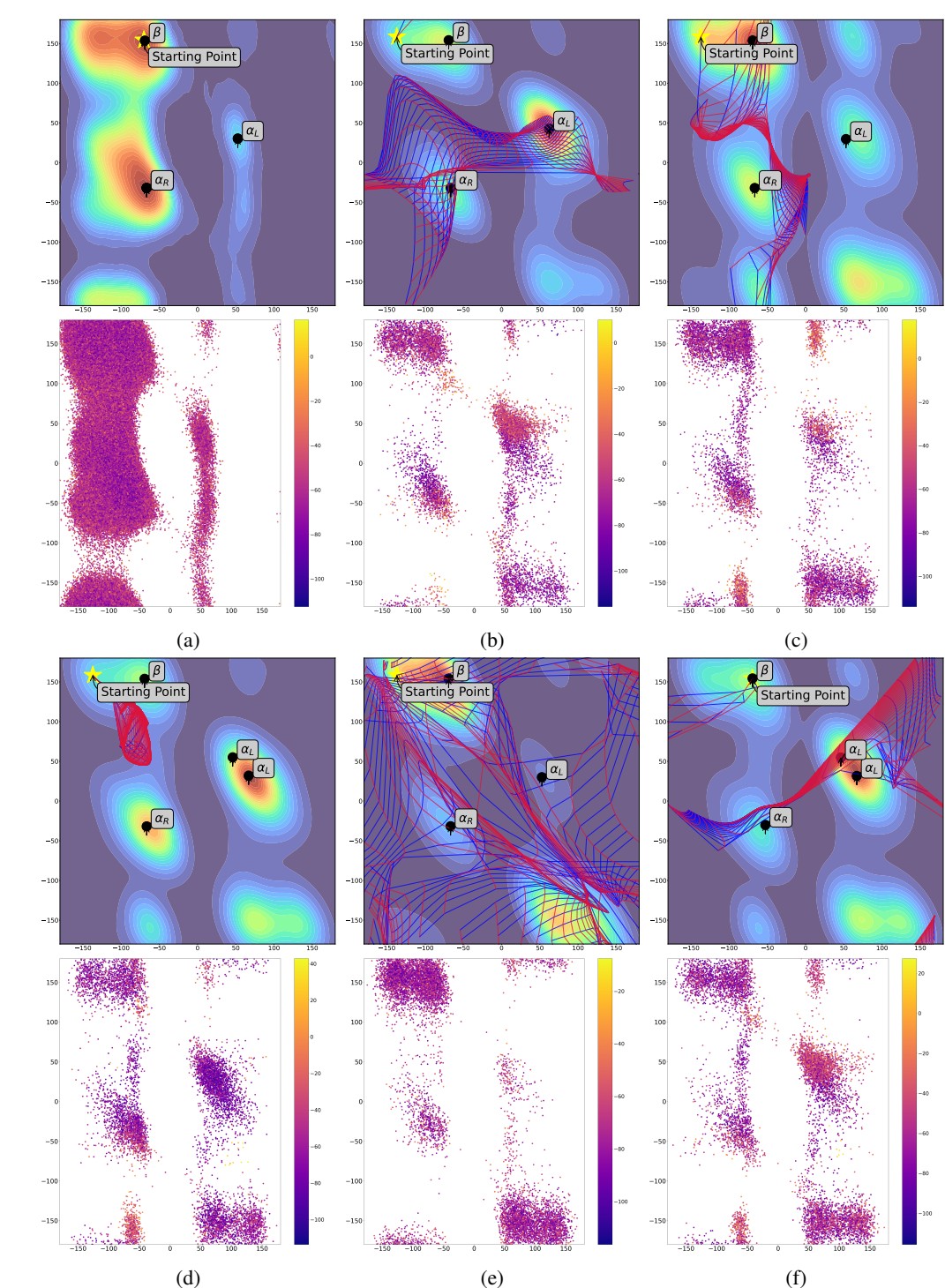

Figure 6: Ramachandran plot for alanine dipeptide in water based on the a) long (500 ns) openMM simulation with implicit solvent starting at $\beta$ alanine dipeptide, b) direct optimization based approach over the molecular forcefield, c) analytically solving the full $\sigma^4$ term in equation 3 d) slow subspace of $H_2$ e) fast degenerate subspace of $H_2$ f) direct optimization based approach over the solvent and the molecular forcefield . The blue and red grid lines on the plots refer to the grid traced by transforming $\beta$ alanine using the two most effective DOF discovered by our algorithms. Additionally the scatter plot gives the values of the potential energy (in presence of solvent) at the points sampled using the corresponding method.

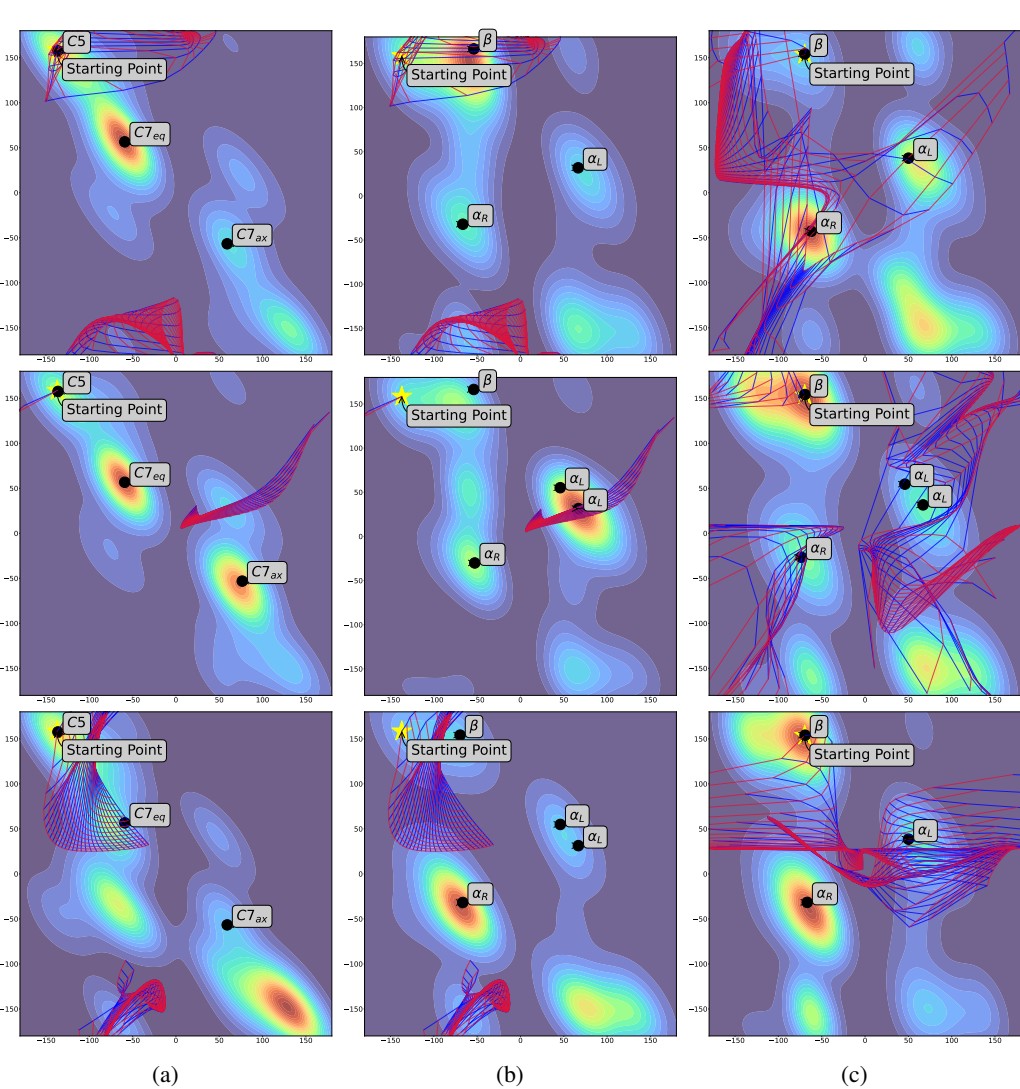

Figure 7: The gridlines and conformers found for three independent runs (row 1, row 2 and row 3) using the direct optimization method with $\varepsilon_1 = 0.1$ and $\varepsilon_2 = 0.01$. a) give the results in vacuum b) gives the results where the simulation is conducted with solvent but the initial trajectory is derived without solvent and c) where the simulation is conducted in solvent and the initial optimization problem is also solved using the solvent forcefield.

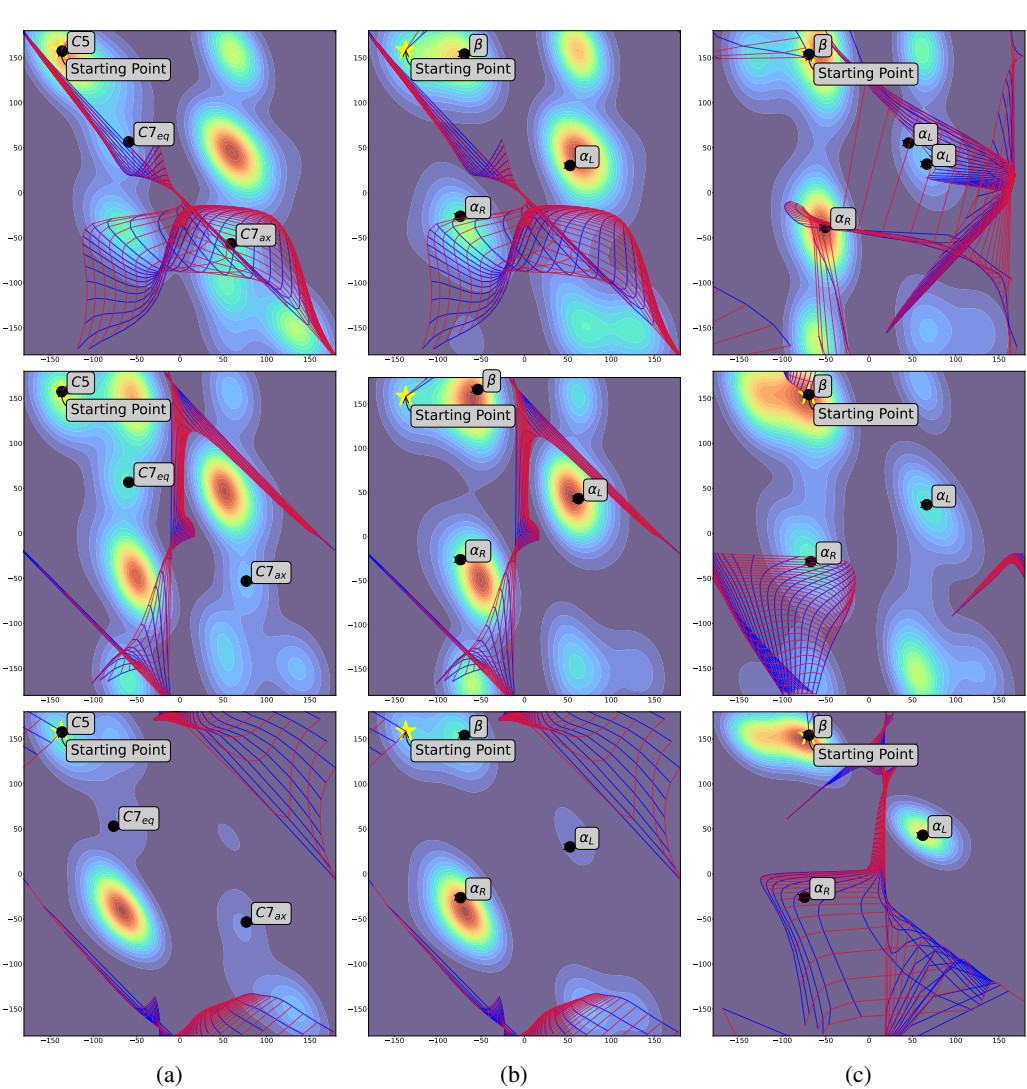

Figure 8: The gridlines and conformers found for three independent runs (row 1, row 2 and row 3) using the direct optimization method with $\varepsilon_1 = 0.5$ and $\varepsilon_2 = 0.01$. a) give the results in vacuum b) gives the results where the simulation is conducted with solvent but the initial trajectory is derived without solvent and c) where the simulation is conducted in solvent and the initial optimization problem is also solved using the solvent forcefield.

