# OpenReview forum: "Symmetry-Driven Discovery of Dynamical Variables in Molecular Simulations"
_ICLR.cc/2025/Conference — Submitted to ICLR 2025_

### Official Review · Reviewer_w5rx · 2024-10-23

**Soundness:** 3
**Presentation:** 2
**Contribution:** 2
**Rating:** 5
**Confidence:** 3

**Summary:**

In this paper, the authors develop two novel approaches for the analysis of degrees of freedom of molecular dynamics simulations. The key idea is to study the symmetry of energy landscape with Lie algebra and further connect the structure dynamics with energy landscape symmetries. Two detailed approaches are considered, including a scalable symmetry loss function and a Hessian-based method. The proposed model has been validated on alanine dipeptide molecular dynamic simulation dataset. The identified two effective degree of freedoms are the two dihedral angles over the peptide bonds, which are consistent with setting.

**Strengths:**

The authors propose two new models for for the analysis of degrees of freedom of molecular dynamics simulations, by the consideration of symmetry of energy landscape. The method are very novel and interesting. The results on the Alanine dipeptide are consistent with the general setting.

**Weaknesses:**

1) The method have only be validated on a "toy"-type example of Alanine dipeptide, which contains only two degrees of freedom. It is not clear the potential performance of the model on more general MD simulations, in which more degrees of freedom are standard cases. Further, the authors have not compare with other models. It is not clear what is the potential advantage of the current models over existing models.
2) The code is not available, thus it is hard to really evaluate the algorithm.
3) Even though this is an interesting algorithm for analyzing MD simulation data, it does not fall into the general machine/deep learning category. I would think the paper is more suitable for a computational chemistry/biology journal.
4) The paper is badly prepared with lots of typos and mistakes. Details will be given in the question part.

**Questions:**

1) The missing of the code and data for the paper. Thus it is impossible to check or reproduce the results in the paper.
2) Too many typos and mistakes, for instance
    a1. Page 1, "a symmetry. loss"
    a2. Page 5, "0 The second term"
    a3. Page  7, "equation ??"
    ....
    In general, the paper is poorly prepared!
3) The advantage of the model and the potential application of models on complicated systems are still not clear. The toy-example of Alanine dipeptide has only two very simple degree of freedom, which should be (easily) identified. What is the advantage of the current model over existing models? How is the potential of using the current model for more realistic complicated MD simulations.

---

> ### Author Response · Authors · 2024-11-21
>
> Thank you for your comments. WE have updated the figures in the paper and are working on fixing typos. Our detailed response:
>
> 1. **Only validated on "toy" example Alanine dipeptide:** While we agree Alanine dipeptide is toy baseline (the MNIST of comp bio), we want to emphasize that, to our knowledge, **no prior work has discovered the torsion angles** in the interpretable way our method does, and certainly not without requiring data. Therefore, we felt that it was absolutely necessary to validate our method on Alanine dipeptide. We do agree that larger systems would be the true test and we are working on longer peptide chains and small proteins. The challenge is that a nice phase space like the one for Ala dipeptide does not exist for these larger systems, so it would take a lot more computational biology discussions and simulations to have presentable results. This would not leave enough room to discuss our methodology and therefore we think larger systems should be a follow-up work.
> 2. **Not clear the potential performance of the model on more general MD simulations, in which more degrees of freedom are standard:** One advantage of our method is that it can discover many DoF easily. In fact, the second step (maximization of symmetry-breaking, eq 15\) is there to sift out the many DoFs and just pick a few. Most steps in our method are very cheap and just require a simple eigendecomposition. For example, using Hessian backbone degenerate subspace, we easily found 6 degenerate vectors. Any pair of these vectors can be used to define an $\\mathbf{L}$ as a rotation in this 2D subspace, yielding 15 (6 choose 2\) DoFs.
> 3. **Not compare with other models…not clear what is the potential advantage over existing models:** We have added a general comment above where we discuss related work. We think that our method provides an orthogonal contribution to the works mentioned above, because:
>    1. **Conceptual leap:** Other works think of DoF (or collective variables) as vectors, while we map them to transformations
>    2. **Data-free:** All other methods require many real or simulated conformations, but our method just needs a decent energy function (forcefield).
> 4. **Code not available:** We have added the code in the Supplementary Materials.
> 5. **Interesting algorithm for analyzing MD, it does not fall into the general machine/deep learning category…more suitable for a computational chemistry/biology journal:** We should have chosen the category more carefully. But, we believe because this is an ML algorithm, an ML venue is much better qualified to judge its validity than a bio venue. Additionally, comp bio, drug discovery and physical simulation are among the fastest growing and most important use cases of ML.
> 6. **Typos and writing quality:** We apologize for this. We are doing our best to polish the paper. We have updated the figures, and will send another note when the typos have been fixed.

---

> > ### Comment · Reviewer_w5rx · 2024-11-25
> >
> > The authors have addressed most of my concerns. However, the key issue still remains, just as the authors mention, " The challenge is that a nice phase space like the one for Ala dipeptide does not exist for these larger systems, so it would take a lot more computational biology discussions and simulations to have presentable results". The potential application and effectiveness of the model on large and "meaningful" systems still remain to be a huge unsolved problem. Thus I will keep my score.

---

> > > ### Author Response · Authors · 2024-11-26
> > > **Clarification**
> > >
> > > We wanted to clarify that this is exactly why our method is useful. For larger systems, finding low energy DoF is challenging, but our model offers a solution. If such DoFs had been worked out for other systems, we would be happy to compare against them.
> > > So our point is that our method can easily be applied to larger systems. It is the baseline and the ground truth that does not exist in the literature for larger systems. Hence we disagree with the statement that  "application and effectiveness of the model on large and "meaningful" systems still remain to be a huge unsolved problem".

---

> ### Author Response · Authors · 2024-11-21
> **Code**
>
> We have now shared the code in the supplement.

---

### Official Review · Reviewer_TfVz · 2024-11-04

**Soundness:** 2
**Presentation:** 2
**Contribution:** 2
**Rating:** 5
**Confidence:** 5

**Summary:**

The others propose a method that tries to identify the effective degrees of freedom of molecular simulations. They do this 2 ways, with Hessian based methods for small systems, and with a loss function for larger ones. They suggest this can explore the conformation space better.

**Strengths:**

They describe interesting theory that indeed is useful for molecular modeling.

**Weaknesses:**

They do not mention existing similar work in machine learning within the last 4 years it seems.
The figures need to be improved. Which is a? b? c? etc in Figure 1 for instance.
The results do not seem to match their analysis. Their only experimental results is Ramachandran plots of alanine dipeptide. They mention that 'we see that almost all the Hessian-based methods recover all the major
conformations of alanine dipeptide with relatively short simulation times.'. but this is very difficult to see from their results. All the plots seem to show completely different distributions.
They do not compare to other baselines in recent literature. (https://pubs.acs.org/doi/10.1021/acs.jctc.4c00454, https://pubs.aip.org/aip/jcp/article/160/17/174109/3287814/Deep-learning-path-like-collective-variable-for,
https://openreview.net/forum?id=TnIZfXSFJAh)
They do not show the distributions of energies of the generated molecules. It could very well be the Ramachandran plots look decent, but the energies are completely off and non-physical. Judging by the molecules they show, it looks like the energies for many of them are very high.

**Questions:**

Can you plot the distribution of energies?
Can you share the github page?

---

> ### Author Response · Authors · 2024-11-21
>
> Thank you for the very pertinent comments. Our response:
>
> 1. **They don’t mention similar work:** We are improving the related work. Please see general comments above.
> 2. **Figs need improvement. Which is a? b? c? etc in Figure 1:** We agree and apologize. We are updating the figures. Please see the updated paper. We have remade all figures:
>    1. Added abc to Figs 1, 3
>    2. Aligned layouts of Figs 2, 4 to make comparison easier
>    3. Fixed a bug in drawing gridlines in Figs 1, 3
>    4. Labeled location of known conformers on the density plots.
>
>    Please also read our summary in the general comment above about what each of our four algorithms discovers.
>
> 3. **Plot the distribution of energies:** Thanks, a very good point. We have added Figs 5, 6 showing the energies of the discovered conformations. Please also see our general comment above titled **Energies of conformations** which shows tables of energies of prominent conformations with or without solvent using each of our methods. In short, our energies are very good. We are adding the tables to the appendix.
> 4. **Code not available:** We have added the code in the Supplementary Materials.
> 5. **\[maybe\] the Ramachandran plots look decent, but the energies are completely off and non-physical:** We should clarify that we take steps to avoid unphysical states. Our procedure is:
>    1. Use discovered DoF to get a starting configuration for short simulations.
>    2. Use openmm optimize to resolve clashes
>    3. Run short openMM simulation (1000 steps)
>
>    It is true that the starting point we get from our DoF often has very high energies because we only look for approximate symmetries and the maximization step (eq 15\) explicitly chooses the DoFs that maximally change the energy (i.e. violate the symmetry condition the most). However, one call of `openMM.minimize` and a short simulation quickly resolves allosteric clashes and yields good physical states. In the energy distributions in Fig 5, 6 we remove the first 100 steps of simulations to avoid the steps where state may have been unphysical.
>
> 6. **They mention that 'we see that almost all the Hessian-based methods recover all the major conformations .. with short simulations.':** Our new Figs 5, 6 now show this more clearly. The distributions shown below each contour plot are the different steps of the 1000 steps (each 1 femtoseconds) openMM simulations, excluding the first 100 steps to remove initial non-physical states. The colors of points show their energies. We observe that all methods, \*except\* Hessian degenerate subspace, explore all conformations, including rare ones not explored by the long simulation (baseline).
> 7. **They do not compare to other baselines in recent literature.** As discussed in our general comment on related work, we feel our method provides an orthogonal contribution to the works mentioned above, because:
>    1. Our main objective is to data-free exploration of the collective variables (CV) space and
>    2. Unlike the given papers where conformation or trajectory between them is known, our task is to discover new conformers using locally-approximated CVs computed only using the atom positions and the local energy function.
>
>    It may be possible to use the points in the long simulation to apply one of the Cv methods in the papers above, but we can say immediately that our method is faster and cheaper than that because it doesn’t require many simulation points or trajectories. Our method could also be used for the tasks discussed in those papers. Future directions can be understanding/sampling transition paths or providing better estimations of the free-energy surface using the afore-mentioned methods.

---

### Official Review · Reviewer_jhsJ · 2024-11-04

**Soundness:** 3
**Presentation:** 1
**Contribution:** 3
**Rating:** 5
**Confidence:** 3

**Summary:**

The authors propose a method to discover effective degrees of freedom in molecular systems, for the aim of fast exploration of low-energy configuration space without prior knowledge of internal coordinates or collective variables. Effective degrees of freedom are directions in configuration space which approximately preserve the energy, and are in this sense viewed as symmetries of the energy. Specifically, degrees of freedom are unit vectors of the Lie algebra of the general linear group of transformations acting on configuration space. The authors propose a symmetry loss to find effective degrees of freedom, which are those along which energy changes minimally, but where conformation changes significantly. The authors refine this symmetry loss with variants using the Hessian of the energy. The proposed method is validated on the model system of alanine dipeptide, where it recovers all important conformers but with dramatically fewer timesteps of molecular dynamics.

**Strengths:**

I find the work exciting because the application of symmetry discovery to enhanced sampling is novel, introduces a new notion for effective degrees of freedom, and empirically appears to dramatically accelerate exploration of molecular configuration spaces. The explanation and derivation of the method is clear and well-paced. The ideas introduced by this work inspire many directions for future work.

I appreciate the extensive discussion of methods for enhanced molecular dynamics sampling.

**Weaknesses:**

There are issues with presentation which hold this submission back from acceptance.

The major issue is that not enough analysis is given to characterize the learned degrees of freedom.
- How does the conformation qualitatively change for a given learned degree of freedom? Do these actually resemble the individual true torsions?
- It is stated that "the degrees of freedom still remain invariant across" conditions in vacuum and solvent - how can this be true if each column of Figs 2 and 4 discovers slightly different conformers?
- What does the beta-sheet conformation look like, and where does it appear on the Ramachandran plot? Where do the rediscovered conformers lie on the plot?
- In Figs 1 and 3, the twisted gridlines differ for each method, and do not appear to cover the entire phi-psi space. How should I interpret this? It is my understanding that the beta conformation should be located at the origin of these grids, but it is not clear to me where that is located.
- Many conformers shown in Fig 4 do not visually resemble the leftmost column. What is the threshold for dihedral angle similarity?

There are numerous minor errors:
- Citation style should have parentheses (\citep) when not using the authors as a noun.
- Inconsistent capitalization of "Lie algebra", "Alanine dipeptide", "DOF"
- Line 50: "symmetry, loss"
- Line 62: "WE show"
- Line 192: should be $x'=gx$
- Line 286: $H_2$ is defined incorrectly
- Line 286: "Then it also minimizes ... can be minimized"
- Line 343: "Effective-ness"
- Line 345: Missing equation number
- Line 363: "m is the number samples"
- Line 377: "Alanine Dipepetide" is misspelled
- Line 388: "direct otimization"
- Lines 754-755: duplicate statements

Experimental setup is not entirely clear. What optimization algorithm was used to minimize losses? How costly is this optimization in terms of number of energy gradient evaluations?

What are the min/max bounds on the 31x31 gridpoints?

**Questions:**

- The method of discovering effective degrees of freedom is stochastic - would multiple trials from the same initial optimized point be expected to converge to similar effective degrees of freedom? How do the discovered degrees of freedom vary as the initial optimized point is varied?
- Given that the proposed method locally searches for directions where energy remains flat, is it correct that the method would not be able to discover new minima that are separated from the initial point by large energy barriers?
- Starting from a single optimized configuration, is it true that the method only requires calculating the Hessian once?
- What, if any, challenges are expected when applying this method to energy functions defined by neural force fields?

An overview figure visualizing the learned effective degrees of freedom as paths on an energy landscape could help to quickly explain what these effective degrees of freedom are.

Since energy calculation is usually the bottleneck in sampling, one way to quantitatively state your speedup is by listing the number of energy, gradient-energy, and Hessian evaluations required by each method.

Another metric, if just focused on sampling, could be the effective sample size (ESS) as measured for Boltzmann generators:

- Klein, L., Krämer, A., & Noé, F. (2024). Equivariant flow matching. Advances in Neural Information Processing Systems, 36.


The related work section "Identifying the DoF" could touch on more of the "collective-variable-free" literature:
- Sipka, M., Dietschreit, J. C., Grajciar, L., & Gómez-Bombarelli, R. (2023, July). Differentiable simulations for enhanced sampling of rare events. In International Conference on Machine Learning (pp. 31990-32007). PMLR.
- Holdijk, L., Du, Y., Hooft, F., Jaini, P., Ensing, B., & Welling, M. (2024). Stochastic optimal control for collective variable free sampling of molecular transition paths. Advances in Neural Information Processing Systems, 36.

---

> ### Author Response · Authors · 2024-11-20
>
> We really appreciate your helpful comments. Detailed response:
>
> 1. **Qualitative conformation:** We are adding visualizations of molecules transformed using our DoF overlaid onto initial conformation. We'll notify when ready.
> 2. **DoF invariant in solvent and vacuum:** We are rewording this. We meant to say that the DoF were transferable to other conditions: We observe the DOFs discovered in vacuum, and for one starting configurations, allowed us to reach various conformations of alanine dipeptide even in the presence of solvent. Note that the DOFs only to provide starting conformer from which we run OpenMM simulations. These starting points covered a wide enough range of the phi-psi angles to allow us to discover all the conformers with short simulations.
> 3. **What does the beta-sheet conformation look like:** It's as in top left of Fig 4 ($\beta$). It is nearly planar, with $\phi,\psi = -150,150$ falling at the upper left corner of the phase plot. We are updating the images and are using a **star** to show the location of the $\beta$ sheet conformer in the phase space. Other conformers are the hotspots of the density plots. For each of the 3D conformers in the old Fig 4 we will put a marker on the phi-psi plot.
> 4. **In Figs 1 and 3 gridlines differ for each method, do not cover the entire phi-psi:** That is correct. Each method has some pros and cons. Also, we noticed we had bug in plotting the gridlines, which is fixed now. Please see the general comment above for detailed findings regarding each method. TLDR; Only the Hessian subspace rotation method covers all of phi-psi, but all other method do a better job at finding more rare conformations.
> 5. **Fig 4 do not resemble the leftmost column:** Note that phi and psi are only the angles of the two bonds around the central “stem” pointing down. These angles need to match, while the rest of the molecule need not. For instance, the methyl groups on the left and right of the molecule can rotate up or down without costing much energy. That being said, we are improving this plot, and experimenting with overlaying the conformers for better comparison. We will update you on this soon.
> 6. **stochastic \- do multiple trials from the same init converge to similar effective DoF?** Note that our Hessian approaches are not stochastic. But regarding our direct symm loss approach, you are right, they do not converge to the same DoF. Nevertheless, moving along them still allowed us to find all known conformers. We are adding a figure to the appendix showing multiple runs of the direct loss and their trajectories in phi-psi. We will reference it in the coming days. The only method that almost perfectly covers the entire phi-psi is the Hessian backbone degenerate subspace.
> 7. **Method locally searches for flat directions, would not be able to discover new minima behind large energy barriers?** Actually, it can. First, our flatness is approximate. More importantly, the second maximization step (eq 14 and 15) actively works against flatness. It finds directions where the gradient increases the most, thereby moving straight toward energy barriers or walls. The way the flat directions should be understood is by considering the hierarchy of forces here. For example, the quadratic bond length and angle potentials introduce directions in the Hessian which are constant and very large. Compared to these large components, directions corresponding to weak van der Waals forces are extremely flat, but eventually show curvature when moving far enough from the current point.
> 8. **Method only requires calculating the Hessian once?** Yes we only compute the Hessian once for the current experiments. However, the experiments can be done in an iterative manner where newly discovered conformers serve as a starting point for computing local DOFs. For our experiments only one round of Hessian computation suffices to find conformers of alanine dipeptide.
> 9. **What, if any, challenges are expected when applied to neural force fields?** In principle, our direct optimization method should directly work with any forcefield, including neural. In this work, we directly use the OpenMM forcefields. Our Hessian-based methods, on the other hand, require the force fields to provide accurate second order information. This might not hold true for ReLU based networks which have 0 Hessian almost everywhere. Our direct optimization approach should still work as it only requires an energy function and the curvature is estimated from perturbed samples instead of analytically.
>
>
> More in next comment

---

> ### Author Response · Authors · 2024-11-20
> **Continued**
>
> 10. **What optimization algorithm? How costly, in terms of number of energy gradients?** Since this system was small, we directly solved the quadratic problem by doing an eigendecomposition on the matrix. For larger systems one can use linear solvers or gradient-based methods. In the end finding Ls is like a linear regression problem with some regularizers. The only subtlety is whether the large gap in the magnitude of gradients in different directions (i.e. hierarchy of interaction strengths) can cause an issue. This is hierarchy certainly exists in our experiments, but it didn’t cause any issues. We suspect this is because we initialized from an already decent conformer, where bond and angles energies were already minimized. In practice any primitive initial layout for the molecule should work, e.g. using RDKit, which also gets bond lengths and angles right. For the Hessian-based method, we require only one Hessian evaluation, while the optimization-based approach needs us to evaluate O(n^2) force values (energy gradients) on points sampled near the initial conformation.
> 11. **What are the min/max bounds on the 31x31 gridpoints?** The transformations learned in our method correspond to Lie algera elements. We get the 31 grid points by repeatedly applying the learned transform on the starting configuration. x\_t \= x\_0 . L^(2\*pi\*(t/31)) where L is the transformation matrix.
>
> 12. **suggested papers:** Thank you very much. We will review them shortly and write about them.

---

> ### Author Response · Authors · 2024-11-21
> **Update on multiple runs of optimization**
>
> We have added Figs 7, 8 to show how the DoF gridlines change when running direct optimization multiple times with different perturbed samples. We show three runs for different setting with or without solvents. Each column shows one setting. Overall, the number of conformers found in each run varies. We will check if increasing the number of samples can make this approach more robust.

---

### Author Response · Authors · 2024-11-20
**Shared response for all reviewers**

Thank you all. We would like to address some common questions prior to the detailed response. We are updating figures, adding new Figs 5 and 6 which also show the energies and distribution of conformers found using each method. We are also making new 3D visualizations of the conformers and send an update once that's ready.

## Pros and cons of each of our algorithms
The core of our methodology consists of two parts: 1) Discover DoFs as approximate symmetries by minimizing symmetry loss; 2) Choose most important DoFs by maximize symmetry-breaking loss. However, in practice we introduce four algorithms, each with its own pros and cons. Let us detail our observations:
   1. **Symmetry Loss direct optimization:** directly minimizes eq (3) $\\mathcal{L} \= \\nabla E \\cdot \\mathbf{L} x$. It needs a number of perturbed samples for the maximization step eq (15). Its gridlines vary significantly depending on the samples. In the new Fig 4, the grid cover all $\phi$ angles, but only a portion of the $\psi$. This method yielded all known conformers, though higher energy conformers were    rarely found.
   2. **Hessian backbone null space (aka “slow modes”):** Generally doesn’t cover a large portion of phi-psi space and remains fairly close to initialization. Nevertheless, simulations starting from these transformed configurations do reach almost all known conformer hotspots.
   3. **Hessian backbone fast degenerate subspace (subspace rotations):** These yield the best grids, covering most of the phi-psi space, but rare conformers are discovered less frequently than the slow subspace.
   4. **Full Hessian (no backbone)**: The full Hessian has four indices (two spatial and two atom indices). In this method, we flatten the Hessian into a matrix by grouping a spatial and atom index together. The resulting DoF have coverage similar to direct symmetry loss.


## Related Work
We thank the reviewers for pointing us to related works on enhanced sampling methods. Here, we give a brief comparison of our work with the referred papers, which we will add to the related works sections of the paper. One key distinction between our method and other works is that we don't require much data. More details:

- **Collective Variables-based methods**: [1], [2] use neural networks to learn CVs derived from neural network-based embeddings of the system descriptors. To learn these embeddings, the authors **require the use of simulation end-points** (bound vs unbound states, or starting conformer and end conformer for transition paths) as well as valid simulation trajectories to learn CVs. Our approach, on the other hand, uses second-order information about the energy landscape to explore the energy landscape, and **discover new conformers without any prior knowledge** of their existence.
- **CV-free methods**: The CV-free methods ([3], [4]) mentioned by the reviewers bias/modify the equations governing the system dynamics to more efficiently sample rare events and transition paths, respectively. In both cases, the system needs to be trained on multiple trajectories in order to learn these biasing potentials. To learn the biasing potential in [3], the bias is parametrized as a combination of small Gaussian biasing potential along the dihedral angles, which uses knowledge of the underlying DoF. Meanwhile, [4]  requires access to high-quality initial samples of transition trajectories to learn the drift for the Schrodinger bridge formulation of the problem. Our proposed method uses Hessian computation to build multiple starting configurations for exploring the CV space without requiring simulation trajectories.

We feel our method provides an orthogonal contribution to the works mentioned above. Our main objective is to efficiently explore the CV space and discover new conformers using locally-approximated CVs computed only using the spatial co-ordinates of the atom positions and the local energy function. The knowledge of these discovered conformers can then be applied to understanding/sampling transition paths or providing better estimations of the free-energy surface using the afore-mentioned methods.

[1] Machine Learning Derived Collective Variables for the Study of Protein Homodimerization in Membrane (https://pubs.acs.org/doi/10.1021/acs.jctc.4c00454).
[2] Deep learning path-like collective variable for enhanced sampling molecular dynamics   (https://pubs.aip.org/aip/jcp/article/160/17/174109/3287814/Deep-learning-path-like-collective-variable-for).
[3] Differentiable simulations for enhanced sampling of rare events. In International Conference on Machine Learning (pp. 31990-32007). PMLR.
[4] Stochastic optimal control for collective variable free sampling of molecular transition paths. Advances in Neural Information Processing Systems, 36.

---

> ### Author Response · Authors · 2024-11-21
> **Energies of conformations**
>
> Energies are in units  $kJ/mol$. The direct optimization method has two $\epsilon$ parameters (units nanometers), one for sampling around the starting point to estimate the Hessian (eqs 5 and 7), and one for the maximization step to choose best $\mathbf{L}$'s (eq 15).
>
> ## Vacuum Results
> The last three columns are for the three prominent conformations.
>
> |Method | $C_5$ |  $C_7^{ax}$ | $C_7^{eq}$ |
> | --------- | --------- | ---------------- | ---------------- |
> |Long Simulation | -80 | -77 | -83 |
> |Direct Optim $\epsilon= (0.01, 0.1)$ |-80 |-84 | -78 |
> |Direct Optim $\epsilon= (0.01, 0.5)$ |-80 | -78| -78 |
> |Full Hessian		| -80 | -78 | -84 |
> |$\mathbf{H}_2$ backbone (slow subspace) | -80 | -84|  -84 |
> |$\mathbf{H}_2$ backbone  (degenerate subspace)| -80 | -78 | -84|
>
> ## Solvent Results
> These are for simulations in water. The last three columns are for the three prominent conformations.
> "Solvent Direct Optim" means using the symmetry loss eq 3 and 5 with the energy function of a forcefield that includes explicit solvent terms (using openMM).
>
> | Method | $\alpha_L$ | $\beta$ | $\alpha_R$ |
> |--------|------------|---------|------------|
> | Long Simulation | -120 | -127 | -128 |
> | Direct Optim $\epsilon= (0.01, 0.1)$ | -105 | -127 | -128 |
> | Direct Optim $\epsilon= (0.01, 0.5)$ | -120 | -127 | -128 |
> | Solvent Direct Optim $\epsilon= (0.01, 0.1)$| -128 | -127 | -120 |
> | Solvent Direct Optim $\epsilon= (0.01, 0.5)$ | -97 | -127 | -128 |
> | Full Hessian | -120 | -127 | -128 |
> | $\mathbf{H}_2$ backbone (slow subspace) | -128 | -127 | -128 |
> | $\mathbf{H}_2$ backbone (degenerate subspace)| -120 | -127 | -128 |

---

> ### Author Response · Authors · 2024-11-21
> **Code in supplement**
>
> Please see the supplement zip file for the code.

---

### Meta-Review · Area_Chair_j5Jo · 2024-12-18

**Metareview:**

This submission develops several new algorithms for calculating or, to use the parlance of the paper, "discovering" the degrees of freedom in a molecular dynamics (MD) simulation. Leveraging symmetries of the configuration space of such a simulation, the authors derive a new symmetry loss function, which is subsequently optimised.

I found this paper to exhibit the following strengths:

- Tackling a new problem domain with strong theoretical underpinnings. Coming from a mathematics background, I enjoyed the exposition of the paper.
- The proposed method does _not_ require any data but just an energy function; this is very elegant.

However, the present manuscript also suffers from shortcomings, specifically:

- The method is only evaluated on a single example (alanine dipeptide), which makes it hard to assess its overall benefits.
- In a similar vein, the method is not compared to other methods. To some extent, this is due to its special "data-free" nature, but the authors mention that, technically, their method could also address other tasks described in the literature such that some baselines would be available.

Overall, this lack of contextualisation makes it hard to assess the merits of the work. While I emphatically agree with the authors about the relevance of the method (and indeed, MD applications are one of fastest-growing areas of ML these days), the current version of the manuscript appears "out-of-scope" at _first_ glance. To counteract this impression, a more comprehensive experimental suite would be required; alternatively, more weight could be given to the optimisation strategies or—potentially—a derivation of approximation schemes based on neural networks (given the proclivity of ML to assume that NNs are great, scalable function approximators).

It is for these reasons that I suggest rejecting the paper for now. I understand that this is not the preferred outcome for the authors, but I believe that a revision of the work, in particular in light of the issues raised with respect to contextualisation, experiments, and comparison partners, would substantially improve the impact and potential audience of the work.

**Additional Comments On Reviewer Discussion:**

The major weaknesses to be addressed concerned (a) the scope of the experimental validation (`w5rx`, `TfVz`), (b) some presentation issues (all reviewers), and (c) potential issues with generalisation performance/applicability (`jhsJ`, `TfVz`). Some of the concerns could be addressed, with authors promising to improve the overall presentation quality, and—I am very grateful for this—providing code in a supplemental file, but the main weakness about the scope (essentially, a single example is being considered) remains unaddressed.

The authors agree that, to some extent, the new method is applied on "toy data," (akin to the MNIST of CompBio) and every reviewer agreed that such an experiment is indispensable. However, for a machine-learning audience, to draw the analogy to MNIST again, it is not sufficient to show that the method works on toy data; at least some slightly more complex scenarios should be considered. These reservations, which preclude an assessment of the generalisation performance, formed the basis of my overall assessment.

---

### Decision · Program_Chairs · 2025-01-22

Reject